# Developmental potential of aneuploid human embryos cultured beyond implantation

Marta N. Shahbazi [1,10,11], Tianren Wang[2,11], Xin Tao[2,11], Bailey A. T. Weatherbee [1,11], Li Sun[2], Yiping Zhan[2], Laura Keller[3], Gary D. Smith[3], Antonio Pellicer[4,5], Richard T. Scott Jr.[6,7✉], Emre Seli [7,8✉] & Magdalena Zernicka-Goetz [1,9✉]

Aneuploidy, the presence of an abnormal number of chromosomes, is a major cause of early pregnancy loss in humans. Yet, the developmental consequences of specific aneuploidies remain unexplored. Here, we determine the extent of post-implantation development of human embryos bearing common aneuploidies using a recently established culture platform. We show that while trisomy 15 and trisomy 21 embryos develop similarly to euploid embryos, monosomy 21 embryos exhibit high rates of developmental arrest, and trisomy 16 embryos display a hypo-proliferation of the trophoblast, the tissue that forms the placenta. Using human trophoblast stem cells, we show that this phenotype can be mechanistically ascribed to increased levels of the cell adhesion protein E-CADHERIN, which lead to premature differentiation and cell cycle arrest. We identify three cases of mosaicism in embryos diagnosed as full aneuploid by pre-implantation genetic testing. Our results present the first detailed analysis of post-implantation development of aneuploid human embryos.

[1] Mammalian Embryo and Stem Cell Group, University of Cambridge, Department of Physiology, Development and Neuroscience, Downing Street, Cambridge CB2 3DY, UK. [2] Foundation for Embryonic Competence, 140 Allen Road, Basking Ridge, NJ 07920, USA. [3] Department of Obstetrics and Gynecology, University of Michigan, 1301 E Catherine St, Ann Arbor, MI 48109, USA. [4] University of Valencia, Department of Paediatrics, Obstetrics and Gynaecology, Av. Blasco Ibanez, 15, Valencia 46010, Spain. [5] IVIRMA Roma, Largo Ildebrando Pizzetti, 1, Rome 00197, Italy. [6] Rutgers-Robert Wood Johnson Medical School, Department of Obstetrics, Gynaecology and Reproductive Science, 125 Paterson Street, New Brunswick, NJ 08901, USA. [7] IVIRMA New Jersey, 140 Allen Road, Basking Ridge, NJ 07920, USA. [8] Yale School of Medicine, Department of Obstetrics, Gynaecology, and Reproductive Sciences, New Haven, CT 06510, USA. [9] Division of Biology and Biological Engineering, California Institute of Technology, 1200 E California Blvd, Pasadena, CA 91125, USA. [10] Present address: MRC Laboratory of Molecular Biology, Francis Crick Avenue, Cambridge Biomedical Campus, Cambridge CB2 0QH, UK. [11] These authors contributed equally: Marta N. Shahbazi, Tianren Wang, Xin Tao, Bailey A. T. Weatherbee. ✉email: richard.scott@ivirma.com; emre.seli@yale.edu; mz205@cam.ac.uk

Human fecundity is remarkably low. It is currently believed that aneuploidy is one of the major limitations of human reproduction, accounting for approximately 50% of early pregnancy losses[1–4]. Moreover, aneuploidy rates are remarkably high in in vitro fertilized human embryos, with up to 50% of embryos diagnosed as aneuploid based on preimplantation genetic testing for aneuploidies (PGT-A)[5–7]. Aneuploidy may lead to implantation failure, miscarriage, as well as congenital defects[8]. Since very little is known about the impact of specific aneuploidies during the early stages of human embryo development, the point at which embryos with aneuploid cells die remains unclear. This lack of knowledge is largely due to the technical challenges in studying human embryo development beyond implantation (day 7)[9], a period that entails intensive growth, morphogenetic remodeling and changes in stem cell identity and fate[10]. Recent technical advances have allowed human embryos to develop beyond day 7 and up to day 12/13 in vitro, in the absence of maternal tissues[11,12]. Embryos cultured in this system recapitulate the major morphological transformations of in vivo developing embryos, such as separation of the inner cell mass (ICM) into the epiblast that will give rise to the embryo proper and the hypoblast that will give rise to the yolk sac, formation of the amniotic and yolk sac cavity, and differentiation of the trophoblast, the tissue that will form the placenta[11,12]. Therefore, this culture method offers an unprecedented opportunity to characterize the early post-implantation development of aneuploid human embryos in vitro.

Here, we aimed to characterize the developmental consequences of specific whole chromosome aneuploidies during human embryo development beyond the blastocyst stage. We restricted our analyses to embryos that reached the blastocyst stage in the appropriate timing to uncover phenotypes of early post-implantation stages. Our results show that monosomy 21 embryos are 10 times more likely to arrest in culture than euploid embryos, and trisomy 16 embryos present a hypoproliferation defect specific of the trophoblast. Mechanistically, studies using human trophoblast stem cells (TSCs) indicate that increased levels of the cell adhesion protein E-CADHERIN, which is located in chromosome 16, lead to cell cycle arrest and premature differentiation. Moreover, we uncovered three cases of mosaicism in embryos that were initially diagnosed by PGT-A as fully aneuploid, highlighting the potential of our human embryo platform to determine the extent of genetic mosaicism, and its influence on post-implantation human development.

## Results

**Preimplantation development of aneuploid human embryos.** We first sought to investigate the impact of specific aneuploidies on human preimplantation development. To this end, we assessed the karyotype of human embryos when they reached the blastocyst stage, as determined by PGT-A. Among 35,171 embryos collected in a single center where PGT-A was offered to all women (avoiding selection bias), we analyzed 9803 embryos that showed single chromosomal aneuploidy, for which 4,712 were autosomal monosomies, 4,717 were autosomal trisomies and 374 were sex chromosome aneuploidies and compared them with 25,368 euploid blastocysts. In our analyses, we included only full chromosomal aneuploidies involving a single chromosome and therefore all diagnosed mosaic or segmental aneuploidies were excluded. Similarly, we excluded complex aneuploidies involving more than one chromosome.

Our analyses revealed that the most common aneuploidies involved chromosomes 15, 16, 21, and 22 (Fig. 1a). Since the time taken to develop to blastocyst stage provides a readout of successful preimplantation development, we categorized the embryos into two groups: (1) embryos that reached the blastocyst stage after 5 days ($n = 13,358$ embryos) and (2) embryos that reached the blastocyst stage only after 6 days in culture ($n = 21,439$ embryos). We found that the proportion of embryos with single chromosomal gain or loss that reached the blastocyst stage only on day 6 was significantly higher than euploid embryos (Fig. 1b). Although most aneuploidies had an effect on the time needed to reach the blastocyst stage, the loss of a copy of chromosome 3 or 6 or the gain of additional copies of chromosomes 6, 8, 11, 12, or 20 resulted in the greatest consequence for development to the blastocyst stage (Fig. 1b; Supplementary Fig. 1). Similarly, the odds ratio of having a better blastocyst expansion score (an indicator of blastocyst quality) at day 5 was lower for embryos with single chromosomal aneuploidy than euploid embryos (Fig. 1c). Monosomic aneuploidies affected the day 5 expansion more severely than trisomic aneuploidies (Fig. 1c).

Trisomy 15 embryos were similar to euploid embryos in their ability to reach the blastocyst stage by day 5 and showed similar expansion scores compared to euploid embryos (Fig. 1b, c). Similarly, trisomy 16, trisomy 21 and monosomy 21 embryos showed minimal developmental delay compared to euploid embryos (Fig. 1b, c). These results indicate that all chromosomal aneuploidies can reach the blastocyst stage, albeit at different times and morphological properties.

**Post-implantation development of aneuploid human embryos.** Next, we analyzed the extent to which particular aneuploidies would lead to developmental differences during early post-implantation stages, as approximately 30% of human pregnancies are estimated to be lost at this stage[8]. To do so, we took advantage of our recently established method to grow human embryos beyond implantation in vitro[11,12] (see "Methods", Fig. 2a). We selected good quality blastocysts based on the morphological Gardner criteria[13], which classifies blastocyst quality based on expansion, from 1 (early blastocyst, with a cavity less than half the embryo volume), to 6 (late blastocyst that has already hatched from the zona pellucida to initiate implantation), and ICM/trophoblast grade, from A (many cells), to C (very few cells). Out of the 163 embryos we thawed, 152 exhibited A- or B-grade, and only 11 exhibited a C-grade ICM and/or trophoblast. After thawing, human blastocysts were first cultured for 24 h to allow recovery and hatching from the zona pellucida. From 163 thawed embryos, 2 embryos were lost during this procedure, and 26 had to be excluded as they did not hatch (Supplementary Table 1). This further decreased the number of C-grade blastocysts used in this study to 6. After hatching, embryos were transferred to the post-implantation in vitro culture (IVC) medium and allowed to develop for 3 days, at which point they were fixed and analyzed. We could distinguish three different categories of embryos based on the development of the embryonic and extra-embryonic lineages, assessed by the expression of specific molecular markers. Embryos in the first category established OCT4 + embryonic epiblast (precursor of the fetus and amnion), GATA6 + extra-embryonic hypoblast (precursor of the yolk sac), and OCT4- GATA6- extra-embryonic trophoblast (precursor of the placenta). We, therefore, conclude that embryos in this category present a normal morphology and maintain the three main lineages of the blastocyst (these embryos were classified as all lineages). The second category of embryos did not establish either epiblast or hypoblast or both (these embryos were termed no ICM). The third category comprised embryos in which development was arrested (these embryos were termed dead/arrested) (Supplementary Table 2).

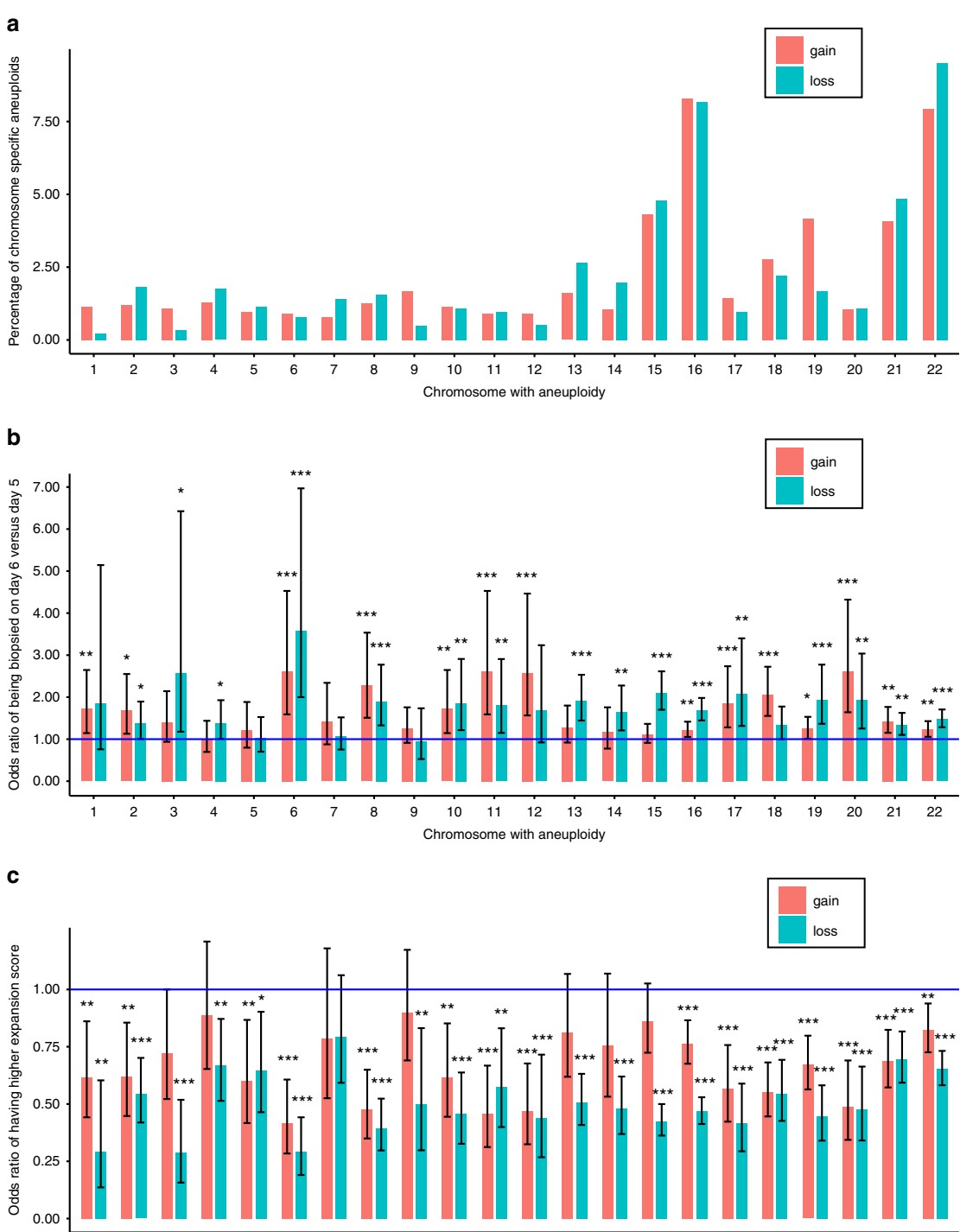

**Fig. 1 Preimplantation development of aneuploid human embryos. a** Prevalence of individual single chromosome aneuploidies among all embryos with single chromosome aneuploidy ($n = 9,429$ embryos). **b** Odds ratios of embryos developing to the blastocyst stage by day 6 rather than day 5 for single chromosome gain or loss ($n = 9,429$ embryos) as compared to euploid embryos ($n = 25,368$). Error bars represent 95% profile likelihood confidence intervals. Blue line represents odds for euploid embryos. Confidence intervals, $p$ values and the specific number of embryos analyzed per genotype is shown in the Source Data file. **c** Odds ratios of embryos having a higher day 5 expansion score for single chromosome gain or loss ($n = 9,429$ embryos) as compared to euploid embryos ($n = 25,368$). Error bars are 95% profile likelihood confidence intervals. Blue line represents odds for euploid embryos. Orange bar represents gain of chromosome (trisomy) while cyan bar represents loss of chromosome (monosomy). Confidence intervals, $p$ values and the specific number of embryos analyzed per genotype is shown in the Source Data file. $*p < 0.05$, $**p < 0.01$, $***p < 0.001$. Source data are provided as a Source Data file.

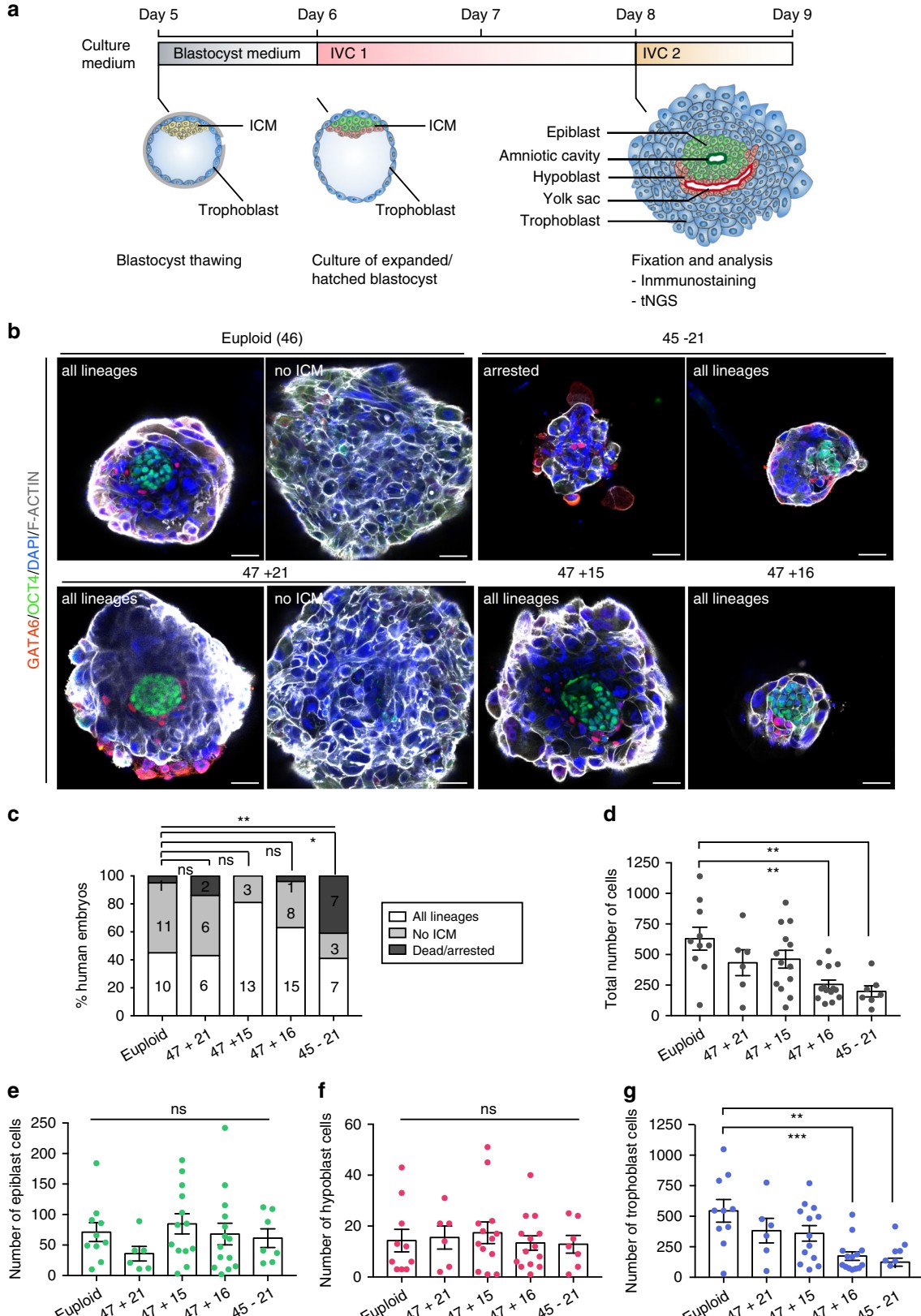

We first analyzed whether in the embryos used in this study the quality of the ICM and/or trophoblast influences early post-implantation development. We focused on A- and B- grade blastocysts, as we did not have enough C-grade blastocysts for comparison. We found that A- and B- grade blastocysts develop similarly up to day 9 (Supplementary Fig. 2a and 2b. n = 54 grade-A and n = 60 grade-B embryos (panel a) and n = 59 grade-A and n = 54 grade-B embryos (panel b)). We also analyzed whether the blastocyst expansion rate has any influence upon subsequent development. While embryos with an expansion score

**Fig. 2 Early post-implantation development of aneuploid human embryos. a** Schematic representation of the methodology used in this study. **b** Immunostaining of human embryos cultured until day 9. Representative images of each karyotype are shown. Scale bars, 50 μm. **c** Developmental phenotypes of embryos from panel (**b**). The number of embryos per category is indicated. Chi-square test, **$p = 0.0018$, *$p = 0.0105$, ns nonsignificant. **d** Total number of cells in embryos with all lineages from panel (**b**). Each dot represents an individual embryo. $n = 10$ euploid, 6 trisomy 21, 13 trisomy 15, 15 trisomy 16 and 7 monosomy 21 embryos. One-way ANOVA with a multiple comparisons test, **$p = 0.0011$ (trisomy 16) and **$p = 0.0027$ (monosomy 21). **e–g**, Number of epiblast (**e**), hypoblast (**f**), and trophoblast (**g**) cells in embryos with all lineages from panel (**b**). Each dot represents an individual embryo, with green for epiblast, red for hypoblast and blue for trophoblast cell counts. $n = 10$ euploid, 6 trisomy 21, 13 trisomy 15, 15 trisomy 16 and 7 monosomy 21 embryos. One-way ANOVA with a multiple comparisons test, **$p = 0.0015$, ***$p = 0.0005$, ns nonsignificant. All error bars represent s.e.m. four independent experiments. ICM inner cell mass, tNGS targeted next generation sequencing, IVC in vitro culture. Source data are provided as a Source Data file.

of 4 and 5 showed a similar developmental potential, embryos with an expansion score of 6 presented a slightly higher incidence of developmental arrest (Supplementary Fig. 2c. $n = 59$ grade 4, $n = 51$ grade 5 and $n = 16$ grade 6 embryos). Previous studies have indicated that implantation and pregnancy rates are higher when embryos reach the blastocyst stage on day 5 compared to embryos establishing the blastocyst stage at day 6[14]. Interestingly, we noted that 75% of embryos with an expansion score of 6 reached the blastocyst stage on day 6 (in comparison with 0% for an expansion score of 4 and 6% for a score of 5). Therefore, we next analyzed whether there were any differences in the developmental competency of day 5 and day 6 blastocysts in vitro. We found that day 6 blastocysts showed more limited development compared to day 5 blastocysts: while 44% of embryos that reached the blastocyst stage at day 5 developed to day 9 with all lineages preserved and presented a normal morphology, none of the embryos that reached the blastocyst stage at day 6 presented a normal morphology at day 9 (Supplementary Fig. 2d and 2e. $n = 16$ day 5 and $n = 14$ day 6 embryos). Based on these findings, we limited all our subsequent analyses to those embryos that reached the blastocyst stage by day 5. The specific criteria used for analysis (hatched day 5 blastocysts with a good embryological score) allowed us to exclude embryos that presented morphological alterations and/or delayed pre-implantation development, although more subtle alterations cannot be excluded.

For further study, we focused on embryos diagnosed as trisomy 21, trisomy 15, trisomy 16 and monosomy 21 as four examples of aneuploidies that are commonly detected at the blastocyst stage (Fig. 1a), have limited impact on pre-implantation development (Fig. 1b, c), and yet lead to very distinct clinical phenotypes. Trisomy 21 is the most common viable trisomy, which affects approximately 1 out of 700 newborns[15]. Trisomy 15 and trisomy 16 typically lead to first trimester miscarriage[16], where most trisomy 16 abortions show empty sacs or minimal development of the embryo, while trisomy 15 miscarriages have specific craniofacial, limb and umbilical cord structural defects[17,18]. We analyzed a total of 71 embryos with these specific aneuploidies: trisomy 21 (14 embryos), trisomy 15 (16 embryos), trisomy 16 (24 embryos), and monosomy 21 (17 embryos) 9 days after fertilization. In addition, we included 22 euploid embryos as controls (Supplementary Table 2). We found blastocysts attached to the dish at day 7-8, which was followed by trophoblast expansion and growth (Supplementary Fig. 3a). We found that attachment occurred irrespective of the genetic makeup of the embryos, with nearly 100% efficiency (Supplementary Fig. 3b). Following embryo development in culture revealed that monosomy 21 embryos exhibited a high rate of arrest by the end of the 9th day of culture, they were smaller in size and contained nuclear debris, indicative of cell death (Fig. 2b, c). These results show that the developmental potential of monosomy 21 embryos is already decreased during the first days of post-implantation development, even in those embryos that reach the blastocyst stage with the

appropriate morphology by day 5. These results are in agreement with a report of defective monosomic embryo attachment in vitro[19], and with the notion that autosomal monosomies lead to pre-clinical pregnancy loss, as they are rarely detected in first-trimester miscarriages[2,4,20,21]. In contrast, we found that over the first days of post-implantation development in vitro, trisomic embryos developed similarly to euploid embryos (Fig. 2b, c). We did not detect any significant differences in the in vitro developmental potential of female and male embryos, either during pre-implantation or early post-implantation development (Supplementary Fig. 4a to 4e. $n = 36$ female and $n = 56$ male embryos).

To characterize, in detail, subtle differences in development beyond day 7, we next focused on the sub-group of embryos in each karyotype that developed with apparently normal morphology and preserved the three lineages of the blastocyst (the first category of embryos: $n = 10$ euploid, $n = 6$ trisomy 21, $n = 13$ trisomy 15, $n = 15$ trisomy 16, and $n = 7$ monosomy 21 embryos). The first morphogenetic transformation of the embryonic epiblast upon implantation is its polarization and epithelialization leading to the formation of the amniotic cavity[9], which can be detected by the presence of the apical protein Podocalyxin (PODXL)[22,23]. We did not observe any significant differences in the efficiency of epithelialization and amniotic cavity formation for the different karyotypes (Supplementary Fig. 5a and 5b). Further analyses of total cell numbers revealed that while trisomy 15 and trisomy 21 embryos did not show any significant differences compared to control euploid embryos, trisomy 16 and monosomy 21 embryos were significantly smaller and had fewer total cells (Fig. 2d). Importantly, this decrease was mainly due to a decrease in the number of trophoblast cells, as we could not detect any significant differences in the number of epiblast and hypoblast cells (Fig. 2e–g). The hypoplastic trophoblast observed in trisomy 16 embryos could potentially explain the intrauterine growth restriction and preeclampsia commonly observed in cases of confined placental mosaicism of trisomy 16[17]. Overall, these results indicate that trisomy 16 and monosomy 21 embryos are already compromised during the first days of post-implantation development.

**Identification of misdiagnosed embryos by PGT-A.** We hypothesized that the specific trophoblast phenotype of some monosomy 21 embryos in the absence of an epiblast and hypoblast phenotype could be due to mosaicism. To test this hypothesis, we manually dissected fixed day 9 embryos into between two and seven pieces based on the size of the embryo, and then lysed, pre-amplified, and assessed chromosome copy number using targeted next-generation sequencing (tNGS). The chromosome copy number results were compared with previous PGT-A results for the same embryo performed on day 5 using either 24 chromosome polymerase chain reaction (PCR) (9 out of 93 embryos), microarray (2 out of 93 embryos), or tNGS (82 out of 93 embryos), as we have previously described[24,25]

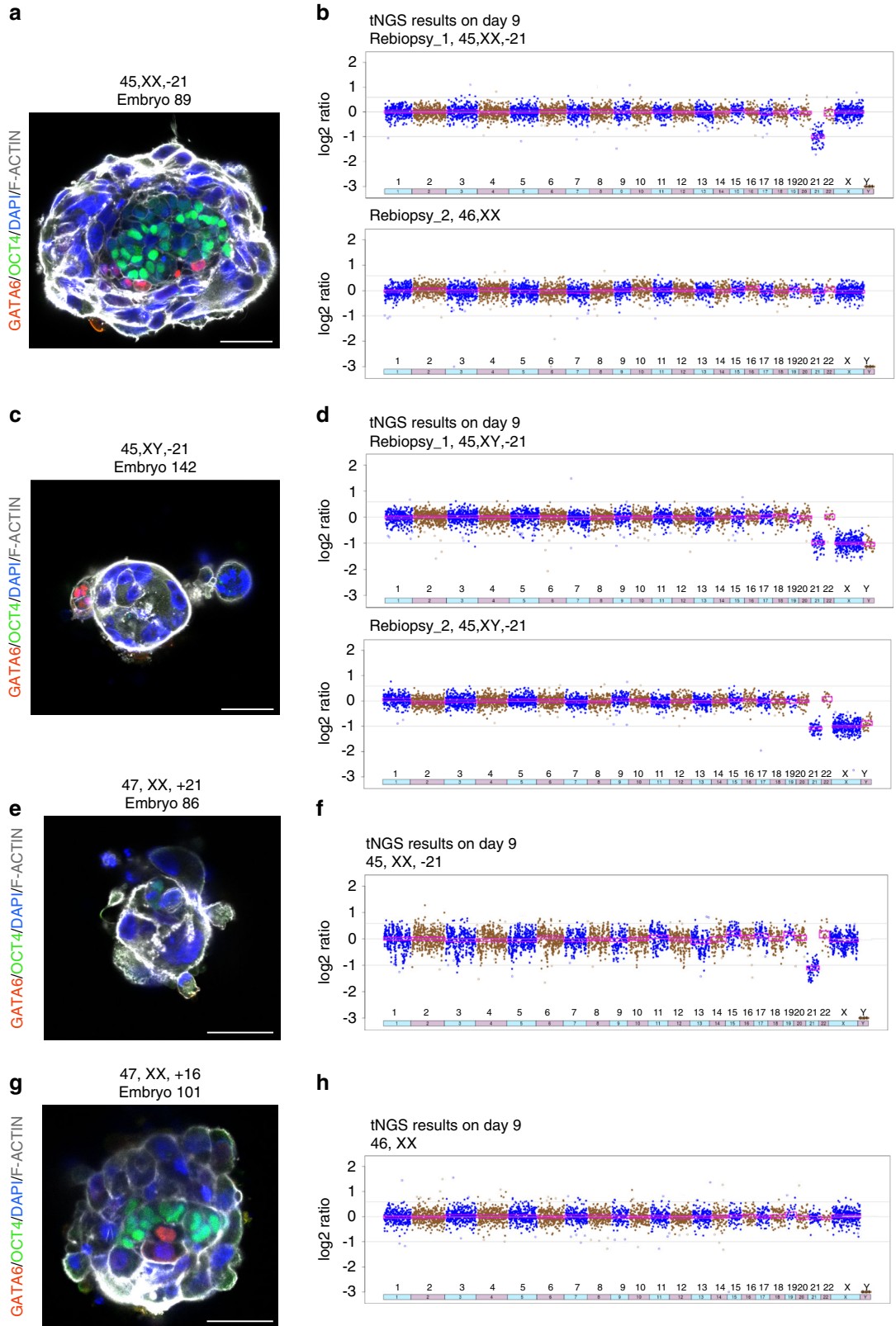

(Supplementary Table 2). The three sections obtained from embryo 89 (identified as 45,XX,−21 based on trophoblast biopsy at day 5 and PGT-A by tNGS) showed 45,XX,−21, 46,XX, and 45, XX,−21 (Fig. 3a, b), confirming the mosaicism for loss of chromosome 21 and indicating the higher developmental potential of mosaic embryos (Fig. 3a), in agreement with previous studies of mosaic mouse and human embryos[19,26,27]. These results are also in agreement with observations indicating that trisomic embryos can give rise to trisomic human embryonic stem cells (ESCs), whereas euploid human ESCs can be obtained from monosomic embryos due to mosaicism[28,29]. Embryo 142, which was diagnosed as 45,XX,−21 based on trophoblast biopsy at day 5 and

**Fig. 3 Chromosome copy number analysis of post-implantation embryos cultured in vitro until day 9. a** Immunostaining of mosaic monosomy 21 embryo (#89, diagnosed as monosomy 21 on day 5) showed normal development of hypoblast and epiblast with limited development of trophoblast. **b** Chromosome copy number analysis from two of the three dissections of the fixed embryo (#89) consistent with mosaic monosomy 21. **c** Immunostaining of arrested monosomy 21 embryo (#142, also diagnosed as monosomy 21 on day 5). **d** Chromosome copy number analysis from the dissections of the fixed dissected embryo (#142) confirmed the previous PTG-A result from day 5 embryo biopsy (45,XX,−21). **e** Immunostaining of a mosaic trisomy 21 and monosomy 21 embryo (#86, diagnosed as trisomy 21 on day 5) showed arrested development. **f** Chromosome copy number analysis showed the presence of monosomy 21 cells. **g** Immunostaining of a euploid day 9 human embryo (#101, diagnosed as trisomy 16 on day 5). **h** Chromosome copy number analysis was consistent with euploidy. All scale bars, 50 μm. tNGS targeted next-generation sequencing.

PGT-A by tNGS, arrested in culture. Chromosome copy number analysis of this embryo on day 9 was consistent with the previous PGT-A result (Fig. 3c, d).

We identified two additional non-concordant cases. Embryo 86 was identified as 47,XX,+21 based on PGT-A by tNGS on day 5, whereas tNGS on day 9 showed a mosaic of trisomy 21 and monosomy 21 cells (Fig. 3e, f), indicative of mitotic nondisjunction. The presence of monosomy 21 cells could explain the arrest of the embryo upon culture to day 9. Embryo 101 was diagnosed as 47,XX,+16 on day 5 by tNGS, and at day 9 was dissected into 6 pieces, all of which proved to be 46,XX, suggesting that the initial PGT-A result could represent a misdiagnosis (Fig. 3g, h). This result could also be explained by selective elimination of the aneuploid cells by programmed cell death as the embryo developed beyond day 5, as previously shown to occur in mouse embryos[26,30]. Alternatively, if the aneuploid cells were restricted to the mural trophoblast these cells could have been removed during the biopsy procedure at day 5. The copy number analyses of the remaining post-implantation embryos (26 out of 29) were consistent with prior PTG-A results (Supplementary Fig. 6a–h).

**Functional characterization of E-cadherin in TSCs.** Next, we wished to explore the mechanism leading to the trophoblast hypoproliferation in trisomy 16 embryos that we detected above. We hypothesized that increased expression of a gene(s) located in chromosome 16 could lead to the above-mentioned phenotype. This turned our attention to the chromosome 16 gene *CDH1*, which encodes the cell-cell adhesion protein E-cadherin (ECAD). ECAD is a transmembrane protein that promotes adhesion between epithelial cells and regulates cell shape[31]. Moreover, by binding intracellularly to the WNT signaling pathway component β-catenin, it modulates the levels of WNT signaling, affecting cellular identity and behavior[32]. To test this hypothesis, we first attempted to derive trisomy 16 human ESCs, with the final aim of differentiating them to trophoblast and decreasing the levels of ECAD. However, it was not possible to derive trisomy 16 human ESC lines from trisomy 16 human blastocysts, while derivation of euploid human ESC lines from euploid blastocysts of equal quality was successful under the same derivation conditions (Supplementary Fig. 7a). As an alternative, and to test whether increased levels of ECAD could lead to trophoblast differentiation and cell cycle arrest, we decided to use human TSCs[33] as a model system. Importantly, human TSCs show transcriptional similarity to early post-implantation cytotrophoblast[34]. In parallel, we used human ESC cultures to model the post-implantation epiblast[35].

We first created stable human TSC and ESC lines that overexpress an ECAD–EGFP fusion protein upon addition of doxycycline (DOX) to the medium (see "Methods"). All of the cell lines were confirmed to be euploid to ensure observed phenotypes were specific to overexpression of ECAD (Supplementary Fig. 7b–e). As a control, we validated that DOX administration in non-transfected cells did not affect proliferation or expression of stemness markers (Supplementary Fig. 8a–g). We then focused our attention on the ECAD–EGFP transfected cells and

performed a time-course analysis of the effects of DOX administration. We found that already after 48 h of DOX addition to the medium, human TSCs transfected with ECAD–EGFP underwent a morphological change, from regular, cuboidal and epithelial, to irregular, flat, and disorganized cells (Supplementary Fig. 9a). After 1 week in the presence of DOX, they stopped proliferating and only a few giant cells could be observed in the dish (Supplementary Fig. 9a). In parallel, we performed equivalent experiments in human ESCs and observed no morphological alterations upon DOX administration (Supplementary Fig. 9a). Importantly, ECAD expression was upregulated in both ESCs and TSCs upon addition of DOX (Supplementary Fig. 9b, c). To characterize this phenotypic change in more detail, we analyzed the levels of the TSC marker GATA3 upon addition of DOX by immunofluorescence. In accordance with the observed morphological change, GATA3 levels significantly decreased upon ECAD–EGFP upregulation (Fig. 4a, b). This was accompanied by an increase in the percentage of multinucleated cells (Supplementary Fig. 9d), and a decrease in proliferation, as indicated by the decreased numbers of mitotic cells labeled by phospho-Histone H3 (pH3) staining (Fig. 4c). Interestingly, ECAD–EGFP upregulation in ESCs did not affect the levels of the pluripotency marker NANOG and only led to a mild decrease in proliferation (Fig. 4e–g). These findings were validated by real-time PCR (RT-PCR); while pluripotency factors NANOG and OCT3/4 did not change upon ECAD–EGFP overexpression in ESCs (Fig. 4h), TSCs overexpressing ECAD–EGFP showed a marked upregulation of the differentiation markers *SDC1* and *HLA-G* (Fig. 4d). We found that the levels of *AXIN2*, a WNT target gene, were significantly decreased in ECAD–EGFP overexpressing TSCs but not ESCs (Fig. 4d, h). To further confirm the effect of ECAD overexpression on TSCs, we sought a second cell type to model the human periimplantation and post-implantation trophoblast. To this end, we converted ESCs that overexpress ECAD in the presence of DOX to TSCs (cTSCs)[36]. Forty-eight hour following addition of DOX to the media, these cells exhibited an increase in *HLA-G* expression and a differentiated morphology (Supplementary Fig. 9e–g). These findings indicate that ECAD overexpression leads to increased differentiation, cell cycle arrest, and decreased WNT activity in human TSCs.

The level of ECAD overexpression with the addition of DOX (1 μg mL$^{-1}$) to TSCs was 200 to 300-fold, which is above the physiological level to be expected with a single additional allele. To address this, we administered lower dosages of DOX and found that a concentration of 10 ng mL$^{-1}$ over 3 days achieved a 1.6-fold overexpression of ECAD (Fig. 5a). At this level of overexpression, the relative expression of differentiation markers *SDC1* and *HLA-G* increased, the levels of the trophoblast marker GATA3 decreased, the proportion of SDC1+ cells and multinucleated cells significantly increased, and the proportion of pH3-positive cells significantly decreased (Fig. 5b–f). This indicates that a physiological upregulation of ECAD is sufficient to induce premature differentiation and cell cycle arrest of human TSCs.

Overexpression of ECAD resulting in increased TSC differentiation was surprising as ECAD expression decreases upon

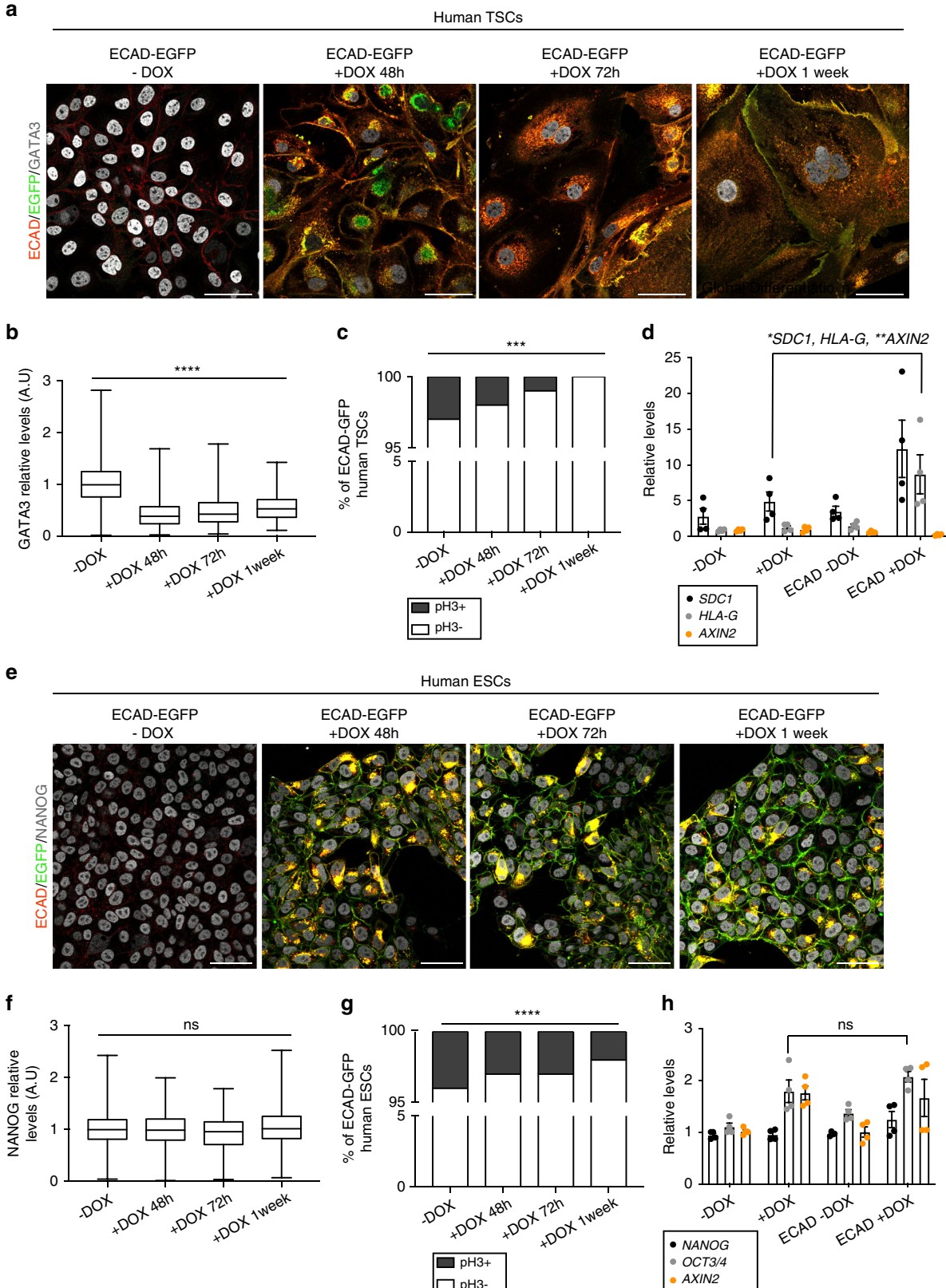

cytotrophoblast differentiation into extravillous trophoblast in vivo[37–39]. We, therefore, asked whether decreasing ECAD levels would be sufficient to affect cell fate. Transient transfection of *CDH1* (ECAD) siRNA resulted in a tenfold decrease in *CDH1* expression compared with control siRNA (Fig. 5g) However, this resulted in no significant difference in expression

of cytotrophoblast markers *GATA3*, *ELF5*, or *TP63* or in expression of differentiation markers *HLA-G* or *SDC1* (Fig. 5h, k). In addition, despite the decrease in ECAD expression, there was no change in *AXIN2* expression, suggesting that the activity of the WNT signaling pathway was unchanged (Fig. 5j). These results suggest that the observed decrease in ECAD expression

**Fig. 4 Characterization of ECAD-overexpressing human TSCs and ESCs. a** Immunostaining of human TSCs transfected with a *CDH1-EGFP* expressing plasmid. ECAD expression is triggered upon 1 µg mL$^{-1}$ DOX addition. **b** Quantification of GATA3 levels in cells from panel (**a**). $n = 2,052, 589, 285,$ and 136 cells per condition. Kruskal Wallis test, ****$p < 0.0001$. Data are shown in a box plot. Whiskers go from minimum to maximum values. The box extends from the 25th to 75th percentile, and the middle line represents the median. **c** Percentage of phospho-HISTONE H3 (pH3) positive cells in cells from panel (**a**). $n = 2,789, 805, 156,$ and 183 cells per condition. Chi-square test, *$p = 0.018$. **d** RT-PCR analysis of *SDC1*, *HLA-G*, and *AXIN2* levels in human TSCs that were/were not transfected with a *CDH1-EGFP* expressing plasmid in the presence or absence of 1 µg mL$^{-1}$ DOX. Each dot represents one sample. $n = 4$ samples per condition. One-way ANOVA with a multiple comparisons test, *$p < 0.0479$, **$p = 0.0022$. Error bars represent s.e.m. **e** Immunostaining of human ESCs transfected with a *CDH1-EGFP* expressing plasmid. ECAD expression is triggered upon DOX addition. **f** Quantification of NANOG levels in cells from panel (**e**). $n = 2,980, 803, 1,080,$ and 1,829 cells per condition. Kurskal Wallis test, ns nonsignificant. Data are shown in a box plot. Whiskers go from minimum to maximum values. The box extends from the 25th to 75th percentile, and the middle line represents the median. **g** Percentage of phospho-HISTONE H3 (pH3) positive cells in cells from panel (**e**). $n = 3,091, 717, 1,497,$ and 2,243 cells per condition. Chi-square test, ****$p < 0.0001$. **h** RT-PCR analysis of *NANOG*, *OCT3/4*, and *AXIN2* levels in human ESCs that were/were not transfected with a *CDH1-EGFP* expressing plasmid in the presence or absence of 1 µg mL$^{-1}$ DOX. Each dot represents one sample. $n = 4$ samples per condition. One-way ANOVA with a multiple comparisons test, ns nonsignificant. All error bars represent s.e.m. three independent experiments (panels **a–c**, **e–g**) and two independent experiments (panels **d** and **h**). All scale bars, 50 µm. Source data are provided as a Source Data file.

upon cytotrophoblast differentiation in vivo may not play a causal role in cell fate determination.

**Trophoblast differentiation in trisomy 16 embryos.** To validate our findings in human TSCs in human embryos, we next cultured in vitro euploid and trisomy 16 embryos up to day 9 and analyzed the levels of ECAD, SDC1 and pH3 in their trophoblast. We found that the trophoblast of trisomy 16 embryos presented increased levels of ECAD, increased SDC1 expression, increased numbers of multinucleated trophoblast cells, and a decrease in the percentage of pH3-positive mitotic cells (Fig. 6a–e. $n = 8$ euploid and $n = 7$ trisomy 16 embryos), in agreement with our previous findings. In summary, our results indicate that the increased levels of ECAD in trisomy 16 embryos contribute to the differentiation and hypoproliferation of the trophoblast compartment without causing major phenotypic changes in epiblast cells at these stages.

## Discussion

Here, we aimed to characterize the development of aneuploid human embryos in vitro with the goal of uncovering the cellular and developmental defects of specific aneuploidies and the molecular mechanisms responsible. To this end, we first compared human blastocysts diagnosed with single aneuploidy to those diagnosed as euploid for their potential for preimplantation development. We found that gain or loss of chromosomes 15, 16, 21, and 22 are most frequently diagnosed in blastocysts, suggesting that development up to the blastocyst stage is more tolerant of errors involving those chromosomes in agreement with previous results[40].

We next focused our attention on particular aneuploidies that are common and cause minimal impact on preimplantation development (trisomy 15, trisomy 16, trisomy 21, and monosomy 21), and selected those human embryos that reached the blastocyst stage with the correct morphology and at the appropriate timing. Even when narrowing down our analysis to this population of good prognosis embryos, our results showed that monosomy 21 embryos are likely to arrest between day 7 and day 9. Upon implantation, the human embryo undergoes a dramatic morphological remodeling, that is associated with an increase in proliferation, cell fate specification events, and lineage commitment[10]. It is therefore likely that embryos lacking an autosome cannot cope with the demands of this developmental phase. Previous reports have shown that monosomic blastocysts graded as high quality already present epigenetic alterations such as hypomethylation, and genetic instability[41], which could contribute to their diminished developmental potential beyond

implantation. Another study analyzed the in vitro attachment of monosomic embryos as a readout of implantation development[19] and found a significant decrease in attachment rates for monosomic embryos. Given that the authors pooled multiple monosomies together, their results may be confounded by the differential phenotypes of specific monosomies. In this regard, we did not detect any significant defect in attachment rates for monosomy 21 embryos.

Surprisingly, a proportion of monosomy 21 embryos survived up to day 9 but displayed a hypoproliferation defect of the trophoblast. Our tNGS results from embryo 89 supports the notion that this phenotype could be a consequence of genetic mosaicism, which was not diagnosed by PGT-A at the blastocyst stage. Given that the current technology does not allow us to separate cells according to their tissue of origin to perform tNGS, it is impossible to conclude whether all ICM-derived cells were euploid and the aneuploid cells were restricted to the trophoblast-derived tissues or mosaicism was also presented in the ICM-derived lineages. In this regard, the fate of aneuploid cells in mosaic human embryos remains unknown. Previous results from our lab have shown that in mouse embryos aneuploid cells in the embryonic epiblast present a higher rate of apoptosis than aneuploid cells in the extra-embryonic tissues[26,30]. These results rely on inducing aneuploidy by the drug reversine, which can cause chaotic aneuploidies[26]. Interestingly, it has been reported that while rates of aneuploidy between the trophoblast and ICM are not significantly different at the blastocyst stage, aneuploid cells are enriched in the trophoblast of in vitro cultured post-implantation human embryos[42]. Moreover, mouse epiblast cells at early post-implantation stages upregulate the expression of proapoptotic genes, which leads to a lower apoptotic threshold in embryonic versus extraembryonic cells in response to DNA damage[43,44]. These studies highlight the different responses of embryonic and extraembryonic cells to damage. However, whether there is a selective elimination of aneuploid cells in the epiblast of post-implantation human embryos remain to be addressed upon development of more advanced methodology.

We also analyzed the development of specific trisomies, namely trisomy 21, trisomy 15, and trisomy 16. Although we did not detect any significant morphological alterations in trisomy 21 and trisomy 15 embryos, whether there are changes at the mRNA and protein levels remains to be explored. In support of this notion, previous reports have shown marked transcriptional alterations in pre-implantation aneuploid human embryos[42,45,46]. Careful analysis of trisomy 16 embryos revealed a marked hypoproliferation phenotype specific to the trophoblast. To study the mechanisms behind this effect, we used human TSCs and ESCs as models of the trophoblast and epiblast respectively. We found

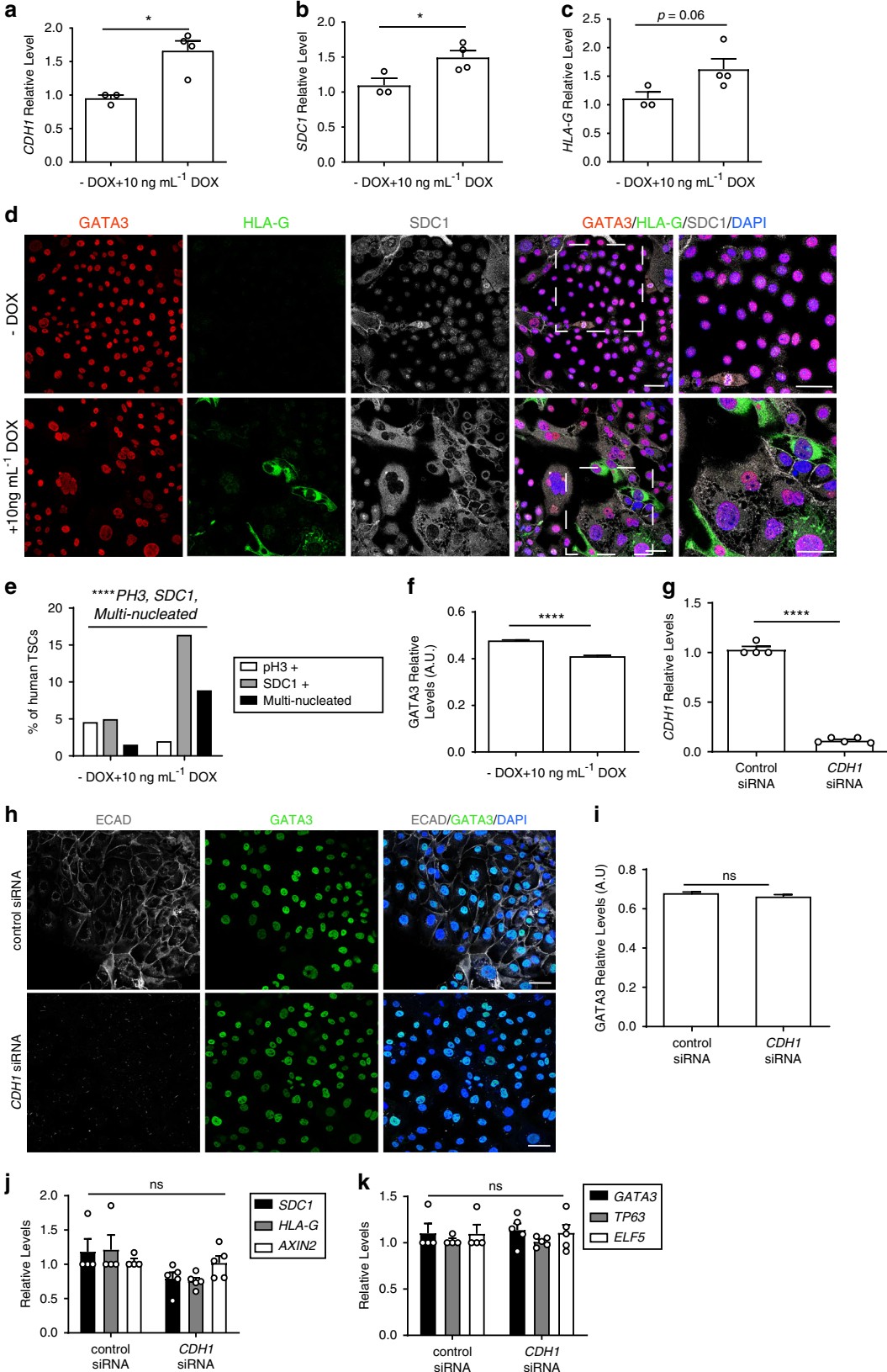

that increased levels of ECAD lead to cell cycle arrest and differentiation in TSCs. The observed effects were specific to TSCs, as ESCs remain pluripotent in the presence of increased levels of ECAD. However, upon conversion of ESCs to TS-like cells, increased levels in ECAD resulted in an upregulation of the

differentiation marker *HLA-G*. In agreement with these findings, a similar phenotype, specific to the trophoblast compartment, was observed in trisomy 16 embryos.

The increased incidence of polyploid cells and the upregulation of SDC1 and HLA-G upon ECAD overexpression are compatible

**Fig. 5 Role of ECAD during trophoblast differentiation. a–c** RT-PCR analysis of *CDH1*, *SDC1*, and *HLA-G* levels in human TSCs transfected with a *CDH1-EGFP* expressing plasmid in the presence or absence of 10 ng mL$^{-1}$ DOX. Each dot represents one sample. $n = 3$ samples for −DOX and 4 samples for 10 ng mL$^{-1}$ DOX. Unpaired Student's *t* test, *CDH1* *$p = 0.0131$. *SDC1* *$p = 0.0357$. *HLA-G* *$p = 0.0634$. **d** Immunostaining of human TSCs transfected with *CDH1-EGFP* expressing plasmid in the presence or absence of 10 ng mL$^{-1}$ DOX. $n = 4$ samples per condition. **e** Percentage of phospho-HISTONE H3 (pH3)-positive cells, SDC1-positive cells, and multinucleated cells in human TSCs transfected with a *CDH1-EGFP* expressing plasmid in the presence or absence of 10 ng mL$^{-1}$ DOX. $n = 743$ and 1,675, 1,407 and 2,752, and 2,132 and 3,707 cells for each condition. Chi-square test, PH3+ ***$p = 0.00321$, SDC1+ ****$p = 0.0001$, multinucleated ****$p < 0.0001$. **f** Quantification of relative GATA3 fluorescence from (**d**). $n = 4,424$ and 2,120 per condition. Unpaired Student's *t* test, ****$p < 0.0001$. **g** RT-PCR of CDH1 in cells transfected with control or *CDH1* siRNA. Each dot represents one sample. $n = 4$ samples for control siRNA and five samples fo *CDH1* siRNA. Unpaired Student's *t* test, ****$p = 0.00000170$. **h** Immunostaining of human TSCs transfected with control or *CDH1* siRNA. **i** Quantification of relative GATA3 levels from panel (**h**). $n = 1,772$ and 1,856 cells per condition. Unpaired Student's *t* test, ns nonsignificant, $p = 0.331$. **j, k** RT-PCR of *SDC1*, *HLA-G*, *AXIN2*, *GATA3*, *TP63*, and *ELF5* in cells transfected with control or *CDH1* siRNA. Each dot represents one sample. $n = 4$ samples for control siRNA and 5 samples fo *CDH1* siRNA. Unpaired Student's *t* test, ns nonsignificant. All error bars represent s.e.m. Scale bars, 50 µm two independent experiments (panels **a**, **b**, **c**, **h**, and **i**) and three independent experiments (panels **d**, **e**, **g**, **j**, and **k**). Source data are provided as a Source Data file.

with the terminal differentiation of TSCs into trophoblast giant cells, although we cannot rule out the possibility that diverse differentiated populations have been specified. WNT signaling is fundamental for TSCs maintenance[33]. In the absence of WNT signaling TSCs differentiate into HLA-G-positive extravillous trophoblast cells[33], precursors of trophoblast giant cells. Interestingly, we observed decreased levels of the WNT target gene *AXIN2* upon ECAD upregulation. These observations suggest that the increased levels of ECAD could sequester β-catenin away from the nucleus (as observed in multiple other systems[47–49]), and therefore lead to a decrease in WNT activity and premature differentiation (Fig. 6f). Overall, our results show that increased levels of ECAD contribute to the hypoproliferation of trisomy 16 trophoblasts. We anticipate that additional changes in protein levels and aneuploid-induced stresses may also contribute to the early lethality of trisomy 16 embryos. To gain further mechanistic understanding of how trisomy 16 affects embryo development we attempted to establish trisomy 16 human ESCs. However, this was not possible, indicating that there may be defects in trisomy 16 epiblast cells preventing robust human ESC derivation. Therefore, whether additional proteins become misregulated in trisomy 16 embryos, especially in the trophoblast compartment, and contribute to the observed phenotype remains unknown.

In summary, our findings show that the IVC method of human embryo culture is a bona fide platform to study early post-implantation developmental competency, a developmental window that is not amenable to study in embryos developing in vivo. Using this system, we have characterized the development of embryos with specific aneuploidies up to day 9 and uncovered tissue-specific alterations. Moreover, our results demonstrate that the in vitro culture platform can be used to identify cases of mosaicism and embryos misdiagnosed by PGT-A. This opens the door for future studies aimed at determining the fate of aneuploid cells during early post-implantation development, and the developmental competency of mosaic embryos, a currently unmet clinical need in human reproduction.

## Methods

**Ethics statement for human embryo experiments**. Human embryos were originally created for purposes of procreation. The study was approved by the Western Institutional Review Board (Clinical IRB 20031397 and 20050731). Additional ethical approval was obtained from the Human Biology Research Ethics Committee (University of Cambridge, HBREC.2017.24). Embryos used in this study were obtained from patients undergoing IVF treatment at IVI-RMA. Informed consent was obtained from all the couples that donated their surplus human embryos.

For human ESC line derivations embryos were donated to the University of Michigan under the category of not suitable for implantation, following PGT-A and/or preimplantation genetic testing for monogenic diseases (PGT-M). Three embryos (UM189-2; UM204-3; and UM230-1) were determined to be Trisomy 16, and the other three embryos (UM161-2; UM178-1; UM207-4) were determined to

be euploid. Written informed consent for donation was obtained as outlined by NIH guidelines, and human ESC line derivation (attempted and/or successful) was performed under University of Michigan's Institutional Review Board approved study, "Derivation of human Embryonic Stem Cells" (HUM00028742).

**Assessment of pre-implantation aneuploid embryos**. All embryos ($n = 35,171$) generated at Reproductive Medicine Associates of New Jersey (RMANJ) in New Jersey, USA between January 2011 and August 2017 were included in the study. A total of 9,429 embryos diagnosed with a single chromosome aneuploidy involving an autosome and 374 sex chromosome aneuploid embryos were compared to 25,368 euploid embryos for their pre-implantation development characteristics. PGT-A (microarray, qPCR or NGS-based, Supplementary Table 2) was carried out at Foundation for Embryonic Competence (FEC, Basking Ridge, NJ, USA) from trophoblast biopsies obtained on day 5 or 6 of in vitro culture, depending on when the embryo reached a developmental stage considered adequate for biopsy. An expansion score (an integer between 0 and 6) was given to each of these embryos on day 5 based on the previously published criteria[13]. A higher score corresponds to an embryo that is more expanded, completely hatched and/or cellular. Logistic regression models were used to assess the odds ratio of embryos being biopsied on day 6 (i.e., developed more slowly) vs. day 5 for each type of single chromosome aneuploidy as compared to euploid embryos. Ordinal logistic regression models were used to obtain the odds ratio of embryos having a higher day 5 expansion score for each type of single chromosome aneuploidy as compared to euploid embryos. Analysis was carried out in R version 3.5.0 with MASS package version 7.3-49.

**Human embryo thawing**. All blastocysts used in this study had undergone assisted hatching on day 3 post-fertilization, trophoblast biopsy after reaching the blastocyst stage on day 5 or 6, PGT-A, and cryopreservation. Embryo quality was assessed prior to embryo freezing using Gardner criteria[13]. Embryos were pipetted using STRIPPER pipettes (Cooper Surgical, US) with disposable plastic tips and thawed using Cryotop Thawing Media Kit (Kitazato, USA). One day prior to thawing the embryos, drops of the human embryo culture medium (Origio Sequential Blast, REF83050010D) covered by mineral oil (Vitrolife, 506061) were incubated in 21% O$_2$/5% CO$_2$ at 37 °C overnight. Dishes (35 mm, BD Falcon, 351008) with prewarmed TS solution and center-well dishes with DS, WS1 and WS2 solutions (1 ml per well) were also prepared prior to starting the thawing procedure. TS solution was prewarmed 2 h in a 37 °C incubator while DS, WS1, and WS2 solutions were prewarmed in room temperature. The straw containing a vitrified blastocyst-stage embryo was removed from the liquid nitrogen and was opened according to manufacturer's instructions. The straw was immediately immersed in the prewarmed TS solution for 1 min; the embryo was located under the dissection microscope and was transferred sequentially to the DS solution for 3 min, WS1 solution for 3 min and WS2 solution for 1 min at room temperature. The embryo was then washed three times in drops of pre-equilibrated complete human embryo culture medium and cultured in a drop of pre-equilibrated complete human embryo culture medium for 24 h.

**Human embryo culture beyond implantation in vitro**. On the first day of the experiment (in vitro culture day 0), IVC1 (Cell guidance system, M11–25) was equilibrated in 21% O$_2$/5% CO$_2$ incubator at 37 °C for a minimum of 1 h, and 300 μL of the pre-equilibrated IVC1 per well was pipetted to an ibiTreat 8-well μ-plate (Ibidi, 80826). The embryo was transferred into the ibiTreat 8-well μ-plate and the plate was immediately placed in a 21% O$_2$/5% CO$_2$ incubator at 37 °C. On the second day of in vitro culture, IVC2 (Cell guidance system, M12–25) was placed in a 21% O$_2$/5% CO$_2$ incubator at 37 °C to equilibrate for a minimum of 1 h; 150 μL of IVC1 medium was removed from the well and 200 μL of pre-equilibrated IVC2 was added. Similarly, until the culture was terminated, medium was changed every day by removing 150 μL and adding 200 μL of pre-equilibrated IVC2 medium.

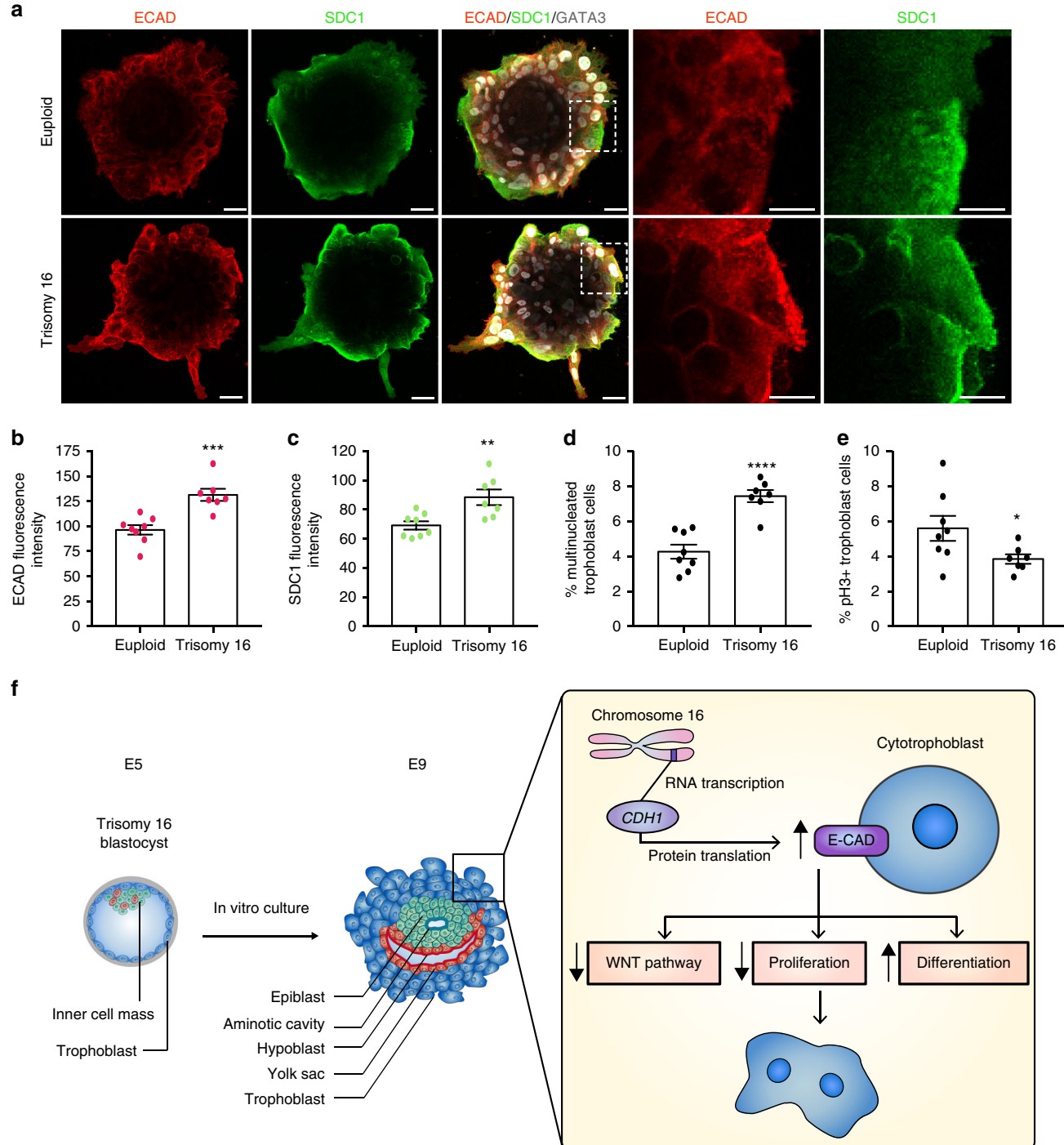

**Fig. 6 Trophoblast characterization in trisomy 16 embryos. a** Immunostaining of euploid and trisomy 16 embryos. Representative images of each karyotype are shown. Squares denote magnified areas. Scale bars, 50 μm. **b**, **c** Quantification of ECAD (**b**) and SDC1 (**c**) levels in embryos from panel (**a**). Each dot represents an individual embryo. $n = 8$ euploid and 7 trisomy 16 embryos. Unpaired Student's *t* test, **$p = 0.0055$, ***$p = 0.0005$. **d** Percentage of multinucleated trophoblast cells in embryos from panel (**a**). Each dot represents an individual embryo. $n = 8$ euploid and 7 trisomy 16 embryos. Unpaired Student's *t* test, ****$p < 0.0001$. **e** Percentage of phospho-HISTONE H3 (pH3) positive cells in embryos from panel (**a**). Each dot represents an individual embryo. $n = 8$ euploid and 7 trisomy 16 embryos. Unpaired Student's *t* test, *$p = 0.0474$. **f** Mechanistic model of trisomy 16 embryo development beyond the blastocyst stage. All error bars represent s.e.m. Source data are provided as a Source Data file.

**Human ESC line derivation**. All human ESC attempted derivations began with vitrified day 5 cryopreserved human blastocyst. Following warming, blastocysts were cultured for 23 h to allow re-expansion followed by microscopic laser-isolation of the ICM of the day-6 blastocysts. Isolated ICMs were plated on inactive Human Foreskin Fibroblasts (HFFs, Global Stem, GSC-3002) in human ESC-Xeno-free Culture Media (Knock-out DMEM (Gibco, 10829) containing 20%

Knockout Serum Replacement (Gibco, 12618012), 1 mM Glutamax (Gibco, 35050-061), 0.1 mM β-mercaptoethanol (Sigma-Aldrich, M6250), 10 mM nonessential Amino Acid 100× (Gibco, 11140-05), and 4 ng ml⁻¹ basic fibroblast growth factor-Xeno-free (MilliporeSigma, GF003AF-100UG). The culture conditions of cell expansion on HFFs were maintained at 5% $CO_2$/5% $O_2$/90% $N_2$ at 37 °C for 5–7 days (passage 0; P0) until a pre-epiblast-like structure was identified

and manually/mechanically split with glass microtools and passaged (P1) onto fresh HFFs, with fresh media, under similar culture conditions. Attempted human ESC derivation was performed with all six human embryos in the same time period, with the same media/additive lot numbers, with the same lot of HFFs, and by the same personnel to reduce inter-attempt variability. For the first 2–5 passages epiblast-like structures and early expanding human ESCs were observed and provided fresh media and additives every other day until manual/mechanical passage onto fresh HFFs (approximately every 7 days). As human ESC lines were established and expanding (P5–7), freeze backs were performed and human ESCs were passaged onto feeder-free matrix, Matrigel (Corning, #354277) with mTeSR1 Media (StemCell Technologies, #85850), with culture conditions of 5% $CO_2$/20% $O_2$/remainder air at 37 °C to allow continued expansion and characterization. Human ESC lines were characterized for pluripotency, genetic composition, multilineage identification (endoderm, mesoderm, and ectoderm) of resulting embryoid bodies, and lack of mycoplasma contamination as indicated for other UM-human ESCs previously produced and published[50].

**Human stem cell culture.** H9 human ESCs were kindly provided by Ludovic Vallier (Stem Cell Institute, UK), under an agreement with WiCell. Experiments with human ESCs were approved by the UK Stem Cell Bank Steering Committee and comply with the regulations of the UK Code of Practice for the Use of Human Stem Cell Lines. CT human TSCs were kindly provided by Hiroaki Okae and Takahiro Arima (Tohoku University Graduate School of Medicine, Japan)[33].

Human ESCs were cultured in matrigel-coated plates in mTESR medium (05825, STEMCELL Technologies). Briefly, plates were coated with 1.6% growth factor-reduced Matrigel (356230, BD Biosciences) dissolved in DMEM/F12 (21331-020, Life Technologies) for 2 h at room temperature. Human ESCs were passaged by a brief treatment with StemPro Accutase Cell Dissociation Reagent (A11105-01, Life Technologies). For the first 24 h after passaging 10 μM ROCK inhibitor Y-27632 (72304, STEMCELL Technologies) was added to the culture. Medium was replaced on a daily basis.

Human TSCs were cultured in collagen-coated plates. Briefly, plates were coated with 0.5% Collagen from human placenta (C7521, Sigma) dissolved in PBS for at least 2 h at 37 C. Human TSC medium contained DMEM/F12 (048-29785, Alpha Labs) or Advanced DMEM/F12 (12634-010, Thermo Fisher Scientific) supplemented with GlutaMAX (35050061, Thermo Fisher Scientific), sodium pyruvate (11360070, Thermo Fisher Scientific), penicillin–streptomycin (15140122, Thermo Fisher Scientific), 100 μM β-mercaptoethanol (31350-10, Thermo Fisher Scientific), 0.2% fetal bovine serum (16141-09, Gibco), 0.3% BSA (017-22231, Wako Chemicals), 1% Insulin–Transferrin–Selenium–Ethanolamine (ITS-X, 51500-056, Thermo Fisher Scientific), 1.5 μg mL$^{-1}$ L-ascorbic acid (A4403, Sigma), 50 ng mL$^{-1}$ human EGF (E9644, Sigma), 2 μM Chiron (produced in house, Stem Cell Institute), 0.5 μM A83-01 (72022, STEMCELL Technologies), 1 μM SB431542 (72232, STEMCELL Technologies), 0.8 mM VPA (227-01071, Wako Chemicals), 5 μM ROCK inhibitor Y-27632 (72304, STEMCELL Technologies). Human TSCs were passaged by treatment with TrypLE Express Enzyme (12604-021, Thermo Fisher Scientific) for 12 min at 37 C. Medium was replaced every other day. Cells were routinely tested for mycoplasma contamination by PCR.

**Conversion of human ESCs.** To convert human ESCs into post-implantation-like TSCs we used an available protocol[36]. Briefly, human ESCs were first plated with mTeSR in wells pre-coated with 5 μg mL$^{-1}$ human recombinant vitronectin (VTN-N; A14700, STEMCELL Technologies) diluted in PBS for 1 h at room temperature. The following three days, cells were cultured in TeSR-E7 media (05914, STEMCELL Technologies) supplemented with 2 μM CYM5541 (4897, TOCRIS), 25 μM SB431542 (72232, STEMCELL Technologies), and 20 ng mL$^{-1}$ BMP4 (120-05, PeproTech), with fresh media changes each day. Next, 50,000 cells were passaged to wells pre-coated with 1 μg Laminin-521 (LN521, Biolamina) and 3 μg mL$^{-1}$ VTN-N diluted in PBS for 1 h at room temperature. Converted TSCs (cTSCs) were cultured in human TSC media as defined above. Cells were passaged 3–5 times in their respective media at a 1:4 ratio, with media change every two days, and then used for experimentation.

**Cloning.** Cloning procedures were carried out using Gateway technology (Thermo Fisher Scientific). A PCR was designed to amplify human *CDH1-EGFP* and to include 5′ and 3′ attB sites using the following oligos:
*CDH1-EGFP* FW: 5′ GGGGACAAGTTTGTACAAAAAAGCAGGCTTC ACCATGGGCCCTTGGAGCC 3′ and *CDH1-EGFP* RV: 5′ GGGGACCACTTTG TACAAGAAAGCTGGGTCTTACTTGATCAGCTCGTCCATGC 3′. This fragment was introduced in a pDONR221 vector (gift of Jose Silva, Stem Cell Institute, UK) using the BP Clonase II (11789020, Thermo Fisher Scientific), and it was further subcloned into a TetO-Zeo plasmid (gift of Jose Silva, Stem Cell Institute, UK) using the LR Clonase II (11791100, Thermo Fisher Scientific).

**Human stem cell electroporation.** Human ESCs and human TSCs were electroporated with three plasmids: CDH1-EGFP-TetO-Zeo, PB-CAG-rtTA3-puto (gift of Jose Silva, Stem Cell Institute, UK) and the pBase plasmid expressing the PiggyBac transposase using the Neon transfection system (Thermo Fisher

Scientific) following the manufacturer's instructions. The following settings were used: for human TSCs 1150 V, 20 ms, 2 pulses; for human ESCs 1200 V, 20 ms, 2 pulses. The following antibiotics for selection were added two days after the transfection: 2 μg mL$^{-1}$ puromycin (ant-pr-1, Invivogen) and 100 μg mL$^{-1}$ zeocin (ant-zn-1, Invivogen). *CDH1-EGFP* expression was triggered by adding 1 μg mL$^{-1}$ or 10 ng mL$^{-1}$ of doxycycline hyclate (D9891, Sigma).

**siRNA-mediated knock-down of *CDH1* in human TSCs.** To knockdown expression of *CDH1* we followed an available protocol for siRNA-mediated loss-of-function experiments in human TSCs[51]. First, a solution of 30 pmol of control or *CDH1* siRNA Silencer Select (4392420, assay ID s531135, Thermo Fisher Scientific), 500 μL of OptiMEM media (31985062, ThermoFisher) and 5 μL of Lipofectamine RNAi MAX (13778075, Thermo Fisher Scientific) per well, was incubated for 10–20 mins at room temperature. Next, 250,000 human TSCs were plated into a 6-well precoated with collagen in 2.5 mL human TSC media. Then, 500 μL of siRNA mix was added. Two days later, this was repeated. Four days after the initiation of siRNA-mediated knockdown, samples were processed for downstream analyses.

**Human embryo and stem cell immunofluorescence staining.** To detect protein expression using immunofluorescence, embryos and stem cells were washed once with phosphate-buffered saline (PBS) and fixed by incubating them in freshly made 4% paraformaldehyde (PFA, 15710, Electron Microscopy Sciences) at room temperature for 20 min. Embryos and stem cells were then washed three times with washing solution at room temperature and then permeabilized by incubating them in either 0.5% (embryos) or 0.3% (stem cells) (vol/vol) Triton X-100 (T8787, Sigma Aldrich) + 0.1 M of glycine (BP381-1, Thermo Fisher Scientific) in PBS at room temperature for 30 min. Then, embryos and stem cells were washed three times with washing solution (0.1% (vol/vol) Tween 20 (P9416, Sigma Aldrich) in PBS) at room temperature for 2 min each and incubated in blocking solution (5% (w/vol) bovine serum albumin (BSA, A9418, Sigma Aldrich) in washing solution) at room temperature for 1 h. Then embryos and stem cells were incubated with primary antibodies diluted 1:200 in blocking solution at 4 °C overnight. After washing the embryos three times in washing solution at room temperature for 2 min, embryos were incubated with fluorescence-conjugated secondary antibodies, Alexa-Fluor®488 Phalloidin (A12379, Thermo Fisher Scientific) and DAPI (D3571, Thermo Fisher Scientific) diluted 1:500 in blocking solution at room temperature for 2 h. Embryos and stem cells were then washed twice in washing solution at room temperature for 2 min and imaged. Primary antibodies: mouse monoclonal anti-ECAD antibody (610182, BD Biosciences, clone 36, 1/100), goat polyclonal anti-GATA3 antibody (AF2605, R&D Systems, 1/200 dilution), goat polyclonal anti-GATA6 antibody (AF1700, R&D Systems, 1/200 dilution), rat monoclonal anti-GFP antibody (GF090R, clone GF090R, Nacalai, USA, 1/1,000 dilution), mouse monoclonal anti-HLA-G (ab7759, clone MEM-G/1, 1/200 dilution), goat polyclonal anti-NANOG antibody (AF1997, R&D Systems, 1/200 dilution), mouse monoclonal anti-OCT3/4 antibody (sc-5279, Santa Cruz Biotechnology, clone C-10, 1/200 dilution), rabbit polyclonal anti-Phospho-Histone H3 antibody (9701, Cell Signaling Technology, 1/200), mouse monoclonal anti-Podocalyxin antibody (MAB1658, R&D, clone 222328, 1/500 dilution), and rabbit monoclonal anti-Syndecan-1 (ab128936, clone EPR6454, Abcam, dilution 1/100). Secondary antibodies: donkey anti-mouse AlexaFluor®568 (A10037, Thermo Fisher Scientific), donkey anti-rabbit AlexaFluor®647 (A31573, Thermo Fisher Scientific), and donkey anti-goat AlexaFluor®488 (A11055, Thermo Fisher Scientific). Stained embryos were imaged by Leica SP8 confocal scanning microscopy.

**Image analysis.** Images were analyzed by Fiji Image J (NIH)[52]. For all quantitative measurements, laser power and detector gain were maintained constant. Cell numbers in embryos were manually counted using the Fiji Cell Counter plugin. Cell numbers in stem cell cultures were automatically counted using the Analyze Particles tool of Fiji. To quantify immunofluorescence levels in embryos a representative single plane capturing the trophoblast was chosen. A region of interest was defined on the outer rim of trophoblast cells. A cytoplasmic and membrane mask was created by subtracting the DAPI signal, which was then used to measure the levels of SDC1 and ECAD. To quantify immunofluorescence levels in stem cells a binarized image of DAPI was used to create a nuclear mask. This mask was then applied to the GATA3 or NANOG channels to measure fluorescence intensity.

**RNA extraction and RT-PCR.** RNA was extracted using TRIzol reagent (15596010, Thermo Fisher Scientific) following the manufacturer's instructions. The reverse transcriptase reaction was performed with 1 μg of RNA in the presence of random primers (C1181, Promega), dNTPs (N0447S, New England BioLabs), RNAse inhibitor (M0314L, New England Biolabs) and M-MuLV reverse transcriptase (M0253L, New England BioLabs). RT-PCR reactions were carried out on a Step One Plus Real-Time PCR machine (Applied Biosystems) using Power SYBR Green PCR Master Mix (4368708, Thermo Fisher Scientific). A list of all the primers used is provided in Supplementary Table 3. The following program was used: 10 min 95 °C denaturation and 40 cycles of 15 s 95 °C and 1 min 60 °C.

**Isolation of genomic DNA from cell lines**. To collect DNA from human ES and TS cell lines for karyotyping, cells from confluent 12- or 6-wells were pelleted and resuspended in 500 µL TNES buffer (50 mM Tris (B2005, Thermo Fisher Scientific) pH 7.4, 100 mM EDTA (15575, Thermo Fisher Scientific) pH 8.0, 400 mM NaCl (27810.262, VWR), 0.5% sodium dodecyl sulfate (SDS)) and 10 µM proteinase K (19131, Qiagen). The solution was incubated at 55 °C for 1 h. Next, 150 µL of 6 M NaCl was added, and the solution was centrifuged for 5 min at full speed at 4 °C. Totally, 500 µL of supernatant was transferred to a new tube and mixed by inverting with 500 µL of ethanol. Next, the solution was centrifuged for 7 mins at full speed at 4 °C. The supernatant was aspirated, and the pellet washed with 200 µL of 70% ethanol. The pellet was airdried for 3–5 min and then resuspended in 50 µL of TE buffer (10 mM Tris pH 8.0, 1 mM EDTA pH 8.0). This was incubated at 65 °C for 20 min or until the pellet dissolved.

**Embryo sample preparation for copy number analysis**. Trophoblast biopsies from blastocysts, or cells dissected from fixed post-implantation embryos were loaded into PCR tubes, and were lysed in alkaline lysis buffer prepared by adding 6 µl molecular biology grade water into PCR tubes, followed by 1 µl alkaline lysis buffer [200 mM KOH and 50 mM DTT]. Samples were incubated at 65 °C for 10 min before 1 µl of neutralization buffer [0.9 M Tris–HCl, pH 8.3, 0.3 M KCl and 0.2 M HCl] was added[53].

**Chromosome copy number analysis by tNGS**. tNGS was performed by the Foundation for Embryonic Competence (FEC, Basking Ridge, NJ, USA). Embryo lysates were amplified using TaqMan Preamplification Master Mix as recommended by the supplier (ThermoFisher Scientific Inc.) in a 50-µL reaction volume with 24 cycles (95 °C for 10 min, 24 cycles of 95 °C for 15 s and 60 °C for 4 min, then 4 °C hold) using an Applied Biosystems 2720 thermocycler, and then quantified with D1k ScreenTape (Agilent Technologies Inc.). Pooled libraries with up to 48 samples were purified utilizing the Agencourt Ampure XP Systems (Beckman Coulter) as per manufacturer recommendations. Ion sphere particles containing clonally amplified libraries were prepared, enriched and loaded to each PI chip using the Ion Chef Instrument (Thermo Fisher Scientific), and then sequenced using the Ion PI Chip V3 and Ion PI Hi-Q Sequencing Kit on the Ion Proton instrument (Thermo Fisher Scientific) following manufacturer's instructions. The reads were filtered for quality and aligned to the human genome, and the copy number of each chromosome was determined[24].

**Chromosome copy number analysis by PCR24**. Multiplex amplification of 96 loci (4 for each chromosome) was performed using TaqMan Copy Number Assays and TaqMan PreAmplification Master Mix as recommended by the supplier (Thermo Fisher Scientific), and in a 50 µl reaction volume for 18 cycles (95 °C for 10 min, 24 cycles of 95 °C for 15 s and 60 °C for 4 min, then 4 °C hold) using a 2720 thermocycler (Thermo Fisher Scientific). Real-time PCR was performed in quadruplicate for each of the individual 96 loci using TaqMan Gene Expression Master Mix (Thermo Fisher Scientific) a 5 µl reaction volume, a 384-well plate, and a ViiA7 real-time PCR system, as recommended by the supplier (Thermo Fisher Scientific). A unique method of the standard delta–delta threshold cycle ($\Delta\Delta C_T$) method of relative quantitation was applied to assess the copy number of individual chromosomes[25].

**Chromosome copy number analysis by SNP microarray**. Whole-genome amplification (WGA) was performed on the lysate according to the manufacturer's instructions starting with library preparation and using the WGA4 GenomePlex Single Cell Whole Genome Amplification kit (Sigma-Aldrich). The GeneElute PCR Purification kit was used to purify WGA DNA from each reaction (Sigma-Aldrich). The WGA or genomic DNA was processed for analysis on the 262 K NspI single-nucleotide polymorphism (SNP) genotyping array as recommended by the supplier (Affymetrix Inc.). Hybridization, washing, staining, and scanning were conducted with the GeneChip Hybridization Oven 640, GeneChip Fluidics Station 450, and GeneChip Scanner 7G, respectively, and as recommended by the manufacturer (Affymetrix). Copy number assignments and loss of heterozygosity analysis results were obtained using the Copy Number Analysis Tool 4.0.1 (Affymetrix). The reference data set consisted of 30 normal female genomic DNA samples.

**Statistical analyses**. Statistical analyses of post-implantation embryo development and stem cell experiments were done in GraphPad Prism. Sample size was determined based on previous experimental evidence. Researchers were not blind to embryo genotype. Qualitative data are presented as a contingency table and were analyzed using a chi-square test. Quantitative data are presented as mean ± s.e.m. with all data points displayed. The normality of the data was analyzed using a D'Agostino-Pearson omnibus normality test, and potential significant differences in the variances were assessed. Data with a Gaussian distribution was analyzed using an unpaired two-tailed Student's $t$ test or a two-tailed ANOVA test with Tukey's multiple comparison test. Data that did not present a Gaussian distribution was analyzed using a two-tailed Kruskal–Wallis test with Dunn's multiple comparison test.

**Reporting summary**. Further information on research design is available in the Nature Research Reporting Summary linked to this article.

## Data availability
All relevant data are available from the authors, with the exception of individual sequencing results that could lead to loss of anonymity for patients who donated their embryos. This is impermissible under approval of the Western Institutional Review Board and patient consents. Source data are provided with this paper.

## Code availability
The codes utilized for chromosome copy number analysis were compiled by the Foundation for Embryonic Competence (FEC), a not-for-profit entity that owns the intellectual property for genome amplification methodology and related analytical code.

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

## Acknowledgements

M.N.S. is funded by the European Molecular Biology Organization (Advanced EMBO fellowship) and by the Medical Research Council (MRC, award MC_UP_1201/24). B.A.T.W. is a recipient of the Gates Cambridge Scholarship. Work in G.D.S. laboratory is funded by an American Society for Reproductive Medicine Research Institute Grant. Work in the M.Z.-G. laboratory is funded by a Wellcome Trust grant (207415/Z/17/Z), Open Philanthropy, Weston Havens, and Curci Foundations.

## Author contributions

M.N.S. helped design the study, performed human embryo, and stem cell experiments and wrote the paper. T.W. ad X.T. designed and performed the human embryo experiments and contributed to writing the paper. B.A.T.W. contributed to designing and performing the stem cell experiments and writing the paper. Y.Z. and L.S. performed the biostatistical analysis and contributed to writing the paper. L.K. and G.D.S. performed the embryonic stem cell derivations. A.P. contributed to the study design and paper. R.T.S., E.S., and M.Z.-G. designed and supervised the study, and wrote the paper.

## Competing interests

A.P. is shareholder of IVI-RMA and DIBIMED. E.S. is a consultant for and receives research funding from the Foundation for Embryonic Competence. The remaining authors declare no competing interests.
