## [Peer Review File · Nature Communications]

Reviewers' comments:

Reviewer #1 (Remarks to the Author):

Shahbazi and colleagues have submitted a paper characterizing the developmental effects of specific whole chromosome aneuploidies using extended in vitro culture of human embryos up to day 9 post fertilization. They demonstrate that embryos harbouring trisomy 21 and 15 developed similarly to euploid embryos. In contrast, embryos with monosomy 21 and trisomy 16 were significantly smaller, mainly due to a decrease in the number of trophoblast cells. Finally, they explore the mechanism leading to trophoblast hypoproliferation in trisomy 16 embryos. Interestingly, they were able to ascribe this to increased levels of the cell adhesion protein E-CADHERIN, which leads to premature differentiation and cell cycle arrest. The authors also identify cases of mosaicism in their day 9 embryos, which were previously diagnosed as uniformly aneuploid. Their technique thus provides an interesting approach for identifying cases of mosaicism and embryos misdiagnosed by PGT-A.

The manuscript is easy and straightforward to read. The design, techniques and sample size are appropriate.

Comments

Pre-implantation development of aneuploid human embryos: correlation with blastocyst Q analysis would also be advantageous

The authors do not report on the quality of blastocysts plated. The authors should incorporate an appropriate embryology grading system into their analysis, to show that the quality of the inner cell mass (ICM) and trophoblast (TE) do not influence post-implantation development. The authors should indicate the quality of all blastocysts used for culture, as well as the quality of the blastocysts that were discarded. Also the number of blastocysts used should be clearly revealed also in the text, although present in the figures and tables, and some statements are formulated based on a low number of embryos which should be taken into account and mentioned.

The authors claim that excluding embryos that reached the blastocyst stage at day 6 allows them to eliminate those alterations arising as a consequence of compromised pre-implantation development, in turn allowing them to solely focus on defects of the second week of development. The authors should acknowledge that they cannot exclude the possibility of all alterations, especially since they do not discuss ICM/TE quality.

In the discussion section the authors discuss their mouse model of chromosomal mosaicism. Although important, the study by Bolton and colleagues is potentially biased, as reversine treatment induces multiple chaotic chromosomal abnormalities. The authors should acknowledge that this model may not be representative of human embryos containing a single mosaic aneuploidy.

Overall, although a major part is more descriptive (and based on a small number of embryos in some parts) and confirming some recent published publications, this study has potential to be valuable for the field of human developmental biology, as well as assisted reproduction, further elucidating the developmental effects of specific whole chromosome aneuploidies.

Specific comments:

- Why the term differentiated was used? "The second category of embryos did not establish either epiblast or hypoblast or both (these embryos were termed "differentiated")."

Reviewer #2 (Remarks to the Author):

The main objective of this study by Shahbazi and colleagues was to examine the consequences of specific chromosomal aneuploidies on early human postimplantation development. Using a previously established in vitro culture system, the authors show that trisomy 15 and 21 embryos develop similar to euploid embryos, whereas monosomy 21 embryos exhibit developmental arrest. They also demonstrate that trisomy 16 embryos exhibit trophoblast hypoproliferation, which the authors claim is due to increased levels of E-CADHERIN, a protein involved in cell adhesion located on chromosome 16. Lastly, the authors suggest that three embryos diagnosed as fully aneuploid by trophoctoderm biopsy and preimplantation genetic screening for aneuploidy (PGT-A) became chromosomally mosaic following postimplantation culture. Overall, the manuscript is fairly well written, but there are multiple issues that need to be addressed in order to support the authors' conclusions of the study as follows:

(1) For the first part of the study, how many embryos were monosomic or trisomic? How many embryos reached the blastocyst stage on day 5 versus day 6? No numbers are provided and it's difficult to extrapolate from the supplementary tables.

(2) The authors state, "By restricting our analyses to embryos that reached the blastocyst stage with a correct morphology and in the appropriate timing..." What is meant by correct morphology? It is fairly well-established that aneuploid blastocysts can appear morphologically normal.

(3) Even though the authors limited their subsequent analyses to hatched day 5 blastocysts "to exclude those alterations arising as a consequence of a compromised pre-implantation development," the embryo phenotypes (trisomies appear normal and monosomies exhibit arrest) are not surprising given several previous publications of similar observations with aneuploid human embryonic stem cells (hESCs) derived from preimplantation blastocysts. Further support is provided by findings that trisomic rescue occurs more frequently than monosomic rescue in human embryos. The study would have been more impactful if they were able to evaluate whether there was selective elimination of aneuploid cells in the epiblast of human embryos as they mention in the Discussion.

(4) This is partially circumvented by the authors' use of hESCs to model postimplantation epiblast and trophoblast stem cells (TSCs) to mimic trophoblast and further study the trisomy 16 trophoblast hypoproliferation, but both cell lines were presumably chromosomally normal. Were they karyotyped? The authors should examine proliferation and differentiation following ECAD knockdown (and overexpression) in trisomic 16 cell lines.

(5) It is stated that, "The chromosome copy number results were compared with previous pre-implantation genetic testing for aneuploidy (PGT- A) for the same embryo performed on day 5 using either 24 chromosome PCR, microarray or tNGS, as we have previously described." How many preimplantation embryos were assessed by each method? The 24 chromosome PCR is not typically used for PGT-A and to my knowledge, has only been tested and validated by one group because which loci were used and the details of the "unique method of the standard delta delta threshold cycle ($\Delta\Delta CT$)" have not been published. These embryos should be removed from the analysis unless they were re-tested by a more comprehensive method.

(6) Also, what was this the method used for the embryos that were mis-diagnosed? Was it 24 chromosome PCR?

(7) How were the TSCs derived, blastocysts or first trimester placentas? No details are provided other than the culture conditions used.

(8) The use of TSCs and hESCs for functional studies are not mentioned in the abstract.

(9) There are some grammatical issues. For instance, see "odds ratio of embryos with single chromosomal aneuploidy to have better blastocysts expansion"

(10) Differentiated is spelled wrong in Figure 5f.

Reviewer #3 (Remarks to the Author):

Overview

In this study, Shahbazi et al., use their recently established in vitro method of culturing human embryos to characterize when and how aneuploid embryos fail during early post-implantation development. First, the authors analyzed a large number of blastocyst embryos tested by PGT-A and found that chromosomes 15, 16, 21 and 22 were most often affected by aneuploidies. Next, Shahbazi et al. confirmed previous findings and validated their culture system by showing that aneuploid embryos reaching the blastocyst stage at day 6 instead of 5, were highly likely to show defects at day 9. This criterion allowed them to focus only on aneuploid embryos reaching the blastocyst stage at day 5 as normal embryos do, and thereby clearly separate defects that arise after implantation from earlier deficiencies. The authors analyzed embryos carrying trisomy 21, trisomy 15, trisomy 16, monosomy 21 and euploid embryos as controls and concluded that: i) monosomy 21 embryos arrested largely by day 9, ii) trisomy 15 and 21 embryos developed normally until day 9, and iii) monosomy 21 and trisomy 16 embryos were significantly smaller due to hypoplastic trophoblast at day 9.

The authors then went on and identified mosaicism in 3 out of 29 embryos with monosomy 21, that accounted for a higher developmental potential of these mosaic embryos. Next, in an attempt to elucidate the mechanism behind the trophoblast hypoproliferation in trisomy 16 embryos, Shahbazi et al., performed some functional studies using human trophoblast stem cells as an in vitro model of trophoblast. They concluded that both in the in vitro model and in the trisomy 16 embryos, increased levels of E-CADHERIN, a gene located on chromosome 16, result in increased differentiation and decreased proliferation of trophoblast cells.

General comments

The number of analyzed embryos in this study is very impressive and the overall characterization will provide the IVF community with an important resource. The general efficiency and quality of the human embryo analysis using the new embryo culture platform is at a high level.

The identification of previously unidentified mosaicism was interesting but did not appear to be further relevant in explaining the phenotype seen in monosomy 21 embryos.

By far the most problematic aspect of the whole paper is the functional analysis using human trophoblast stem cells and the conclusions drawn from this. Firstly, the human trophoblast stem cells are in vitro representation of the villous cytotrophoblast, a cell type that appears later during development. Secondly, Chromosome 16 contains about 800-900 genes (depending on the counting method), of which many are essential for cell viability and during development. Therefore, the hypothesis that the observed trophoblast phenotype is caused by an expected 1.5x upregulation of only one gene (CDH1) appears highly unlikely. However unlikely, this hypothesis could be tested in human trophoblast stem cells if the same gene dosage overexpression of expected 1.5x could be

achieved. Unfortunately, the observed dox-inducible overexpression level of CDH1 seen in Supplementary Figure 8c is instead around 200-300 and therefore of a completely different order of magnitude. Consequently, it is completely unsound to compare these two models and draw any reliable conclusions from experiments in Figure 4 and Figure 5.

Conclusion

This study provides new insights into post-implantation development of human aneuploid embryos and represents an important and useful resource for the IVF community. However, the functional analysis implicating E-CADHERIN as a cause of the trisomy 16 phenotype is deeply flawed and therefore inconclusive.

RESPONSES TO REVIEWERS COMMENTS

Reviewer #1 (Remarks to the Author):

Shahbazi and colleagues have submitted a paper characterizing the developmental effects of specific whole chromosome aneuploidies using extended in vitro culture of human embryos up to day 9 post fertilization. They demonstrate that embryos harbouring trisomy 21 and 15 developed similarly to euploid embryos. In contrast, embryos with monosomy 21 and trisomy 16 were significantly smaller, mainly due to a decrease in the number of trophoblast cells. Finally, they explore the mechanism leading to trophoblast hypoproliferation in trisomy 16 embryos. Interestingly, they were able to ascribe this to increased levels of the cell adhesion protein E-CADHERIN, which leads to premature differentiation and cell cycle arrest. The authors also identify cases of mosaicism in their day 9 embryos, which were previously diagnosed as uniformly aneuploid. Their technique thus provides an interesting approach for identifying cases of mosaicism and embryos misdiagnosed by PGT-A.

The manuscript is easy and straightforward to read. The design, techniques and sample size are appropriate.

We thank the reviewer for his/her appreciation of the quality of our manuscript.

Comments

Pre-implantation development of aneuploid human embryos: correlation with blastocyst Q analysis would also be advantageous. The authors do not report on the quality of blastocysts plated. The authors should incorporate an appropriate embryology grading system into their analysis, to show that the quality of the inner cell mass (ICM) and trophectoderm (TE) do not influence post-implantation development. The authors should indicate the quality of all blastocysts used for culture, as well as the quality of the blastocysts that were discarded. Also the number of blastocysts used should be clearly revealed also in the text, although present in the figures and tables, and some statements are formulated based on a low number of embryos which should be taken into account and mentioned.

We thank the reviewer for this comment, as it has helped us to clarify the relation between embryo quality on days 5/6 and 9. Blastocysts used in this analysis were selected based on their morphology. Most of the embryos had a 4BB score or above (as show in Supplementary Table 3). New Supplementary Figure 2a-c show the influence of blastocyst expansion grade, ICM quality and TE quality on post-implantation development. Embryos with an expansion score of 4 and 5 have similar developmental potential in vitro, while an expansion score of 6 is associated with slightly higher rates of arrest in culture. This could be due to the fact that 75% of embryos in this category were biopsied at day 6 (in comparison with 0% for an expansion score of 4 and 6% for a score of 5). In terms of ICM and TE quality we could not observe any significant differences between A- and B- grade embryos. Therefore, in the blastocysts used in this study, blastocoel expansion, ICM, and TE quality do not influence post-implantation development.

Following the suggestion of the reviewer we now include n numbers in the text for all the experiments.

The authors claim that excluding embryos that reached the blastocyst stage at day 6 allows them to eliminate those alterations arising as a consequence of compromised pre-implantation development, in turn allowing them to solely focus on defects of the second week of development. The authors should acknowledge that they cannot exclude the possibility of all alterations, especially since they do not discuss ICM/TE quality.

As mentioned above we have now analysed the effect of ICM/TE quality in the human embryos used in this study and found no differences between A- and B-grade blastocysts. Nonetheless, we agree with the reviewer that we cannot claim we have eliminated all alterations arising as a compromised pre-implantation development. Therefore, we have:

- Removed our claim of the phenotypes been post-implantation-specific in the introduction.
- Mention that in morphologically normal blastocysts we cannot exclude subtle alterations (results section).
- Comment on the presence of epigenetic alterations in monosomic embryos graded as high quality (discussion section).

In the discussion section the authors discuss their mouse model of chromosomal mosaicism. Although important, the study by Bolton and colleagues is potentially biased, as reversine treatment induces multiple chaotic chromosomal abnormalities. The authors should acknowledge that this model may not be representative of human embryos containing a single mosaic aneuploidy.

We agree with the reviewer and we now include this in our discussion.

Overall, although a major part is more descriptive (and based on a small number of embryos in some parts) and confirming some recent published publications, this study has potential to be valuable for the

field of human developmental biology, as well as assisted reproduction, further elucidating the developmental effects of specific whole chromosome aneuploidies.

We thank the reviewer for his comment but would like to emphasise that in this study we have analysed an unprecedented number of human embryos. We have analysed 35,171 pre-implantation embryos and we have thawed a total of 163 embryos for post-implantation culture. For our analysis of specific aneuploidies, we used from 14 to 24 embryos per group. Additionally, for our detailed analysis of proliferation, differentiation and E-Cadherin expression in trisomy 16 embryos we analysed 8 euploid and 7 trisomy 16 embryos. This is the experiment with the smallest embryo numbers per group in our manuscript.

Specific comments:

- Why the term differentiated was used? "The second category of embryos did not establish either epiblast or hypoblast or both (these embryos were termed "differentiated")."

We agree that the term "differentiated" is not clear and now refer to the second category of embryos as "no ICM".

Reviewer #2 (Remarks to the Author):

The main objective of this study by Shahbazi and colleagues was to examine the consequences of specific chromosomal aneuploidies on early human postimplantation development. Using a previously established in vitro culture system, the authors show that trisomy 15 and 21 embryos develop similar to euploid embryos, whereas monosomy 21 embryos exhibit developmental arrest. They also demonstrate that trisomy 16 embryos exhibit trophoblast hypoproliferation, which the authors claim is due to increased levels of E-CADHERIN, a protein involved in cell adhesion located on chromosome 16. Lastly, the authors suggest that three embryos diagnosed as fully aneuploid by trophoctoderm biopsy and preimplantation genetic screening for aneuploidy (PGT-A) became chromosomally mosaic following postimplantation culture. Overall, the manuscript is fairly well written, but there are multiple issues that need to be addressed in order to support the authors' conclusions of the study as follows:

We thank the reviewer for his/her appreciation of our manuscript.

(1) For the first part of the study, how many embryos were monosomic or trisomic? How many embryos reached the blastocyst stage on day 5 versus day 6? No numbers are provided and it's difficult to extrapolate from the supplementary tables.

We have included these numbers in the text. In addition, we include a new supplementary table (new Supplementary Table 1) that specifies the number of embryos analysed per genotype.

(2) The authors state, "By restricting our analyses to embryos that reached the blastocyst stage with a correct morphology and in the appropriate timing..."What is meant by correct morphology? It is fairly well-established that aneuploid blastocysts can appear morphologically normal.

We agree that we cannot rule out the possibility that other alterations are present in morphologically normal blastocysts. We have revised this sentence and specifically mention that we restrict our analysis to embryos that reach the blastocyst stage in the appropriate timing.

(3) Even though the authors limited their subsequent analyses to hatched day 5 blastocysts "to exclude those alterations arising as a consequence of a compromised pre-implantation development," the embryo phenotypes (trisomies appear normal and monosomies exhibit arrest) are not surprising given several previous publications of similar observations with aneuploid human embryonic stem cells (hESCs) derived from preimplantation blastocysts. Further support is provided by findings that trisomic rescue occurs more frequently than monosomic rescue in human embryos. The study would have been more impactful if they were able to evaluate whether there was selective elimination of aneuploid cells in the epiblast of human embryos as they mention in the Discussion.

Determining the fate of aneuploid cells in mosaic human embryos would be a key achievement for human reproduction. However, with the current technology it is impossible to address this question, as we now mention in the revised discussion. We would need a technology that would allow us to determine which cells are aneuploid in a blastocyst, mark those cells in a non-invasive way, and follow them beyond the blastocyst stage. To the best of our knowledge such technology does not exist. Since the objective of our study was to characterize the early post-implantation phenotype of aneuploid human embryos and determine the molecular mechanisms behind, we believe that following the fate of aneuploid cells in mosaic embryos is beyond the scope of our work.

(4) This is partially circumvented by the authors' use of hESCs to model postimplantation epiblast and

trophoblast stem cells (TSCs) to mimic trophoblast and further study the trisomy 16 trophoblast hypoproliferation, but both cell lines were presumably chromosomally normal. Were they karyotyped? The authors should examine proliferation and differentiation following ECAD knockdown (and overexpression) in trisomic 16 cell lines.

We thank the reviewer for raising this comment, as karyotyping our stem cell lines is an important control. As requested, we have performed NGS analysis of H9 human ESCs, H9 human ESCs overexpressing ECad, CT human TSCs, and CT human TSCs overexpressing ECad. The results show that the four stem cell lines are euploid. These results are shown in the new Supplementary Figure 7

We have tried to obtain trisomic 16 cell lines as an additional way to address this question, but this has proven to be extremely difficult. Trisomy 16 human TSCs do not exist, and based on our results we believe it would not be possible to derive them as the trisomy 16 trophoblast presents a strong proliferation defect.

As a way to circumvent this, we have tried to derive trisomy 16 human ESCs, with the idea of subsequently differentiating these cells to a TSC fate. There are two trisomy 16 human ESC lines described, one by the group of Nissim Benvenisty (Biancotti et al., 2010) and one by the group of Gary Smith (human ESC line UM90-12 PGS available at the NIH stem cell bank). The trisomy 16 line derived by Biancotti et al was lost, potentially due to genetic mosaicism in the culture (Prof. Nissim Benvenisty, personal communication). Trisomy 16 line UM90-12 PGS is also mosaic (Prof. Gary Smith, personal communication). For this reason, we have attempted to establish new trisomy 16 human ESC lines in collaboration with Prof. Gary Smith. As can be seen in the attached figure we have used 3 trisomy 16 blastocysts and 3 control embryos. The control blastocysts gave rise to human ESC colonies. However, while the trisomy 16 embryos initially attached, and formed an outgrowth, they eventually degenerated. This suggests that while the epiblast of trisomy 16 embryos can initially proliferate *in vitro*, the cells eventually die as a consequence of the extra chromosome 16. Therefore, we are technically unable to address the comment of the reviewer. Nonetheless, as a control we have performed a knock-down of ECAD in wild type TSCs and found that a decrease in basal ECAD levels does not affect TSC fate. Therefore, the phenotype we report is specific to ECAD overexpression.

Figure: attempted derivations of trisomy 16 human ESCs.

(5) It is stated that, “The chromosome copy number results were compared with previous pre-implantation genetic testing for aneuploidy (PGT- A) for the same embryo performed on day 5 using either 24 chromosome PCR, microarray or tNGS, as we have previously described.” How many preimplantation embryos were assessed by each method? The 24 chromosome PCR is not typically used for PGT-A and to my knowledge, has only been tested and validated by one group because which loci were used and the details of the “unique method of the standard delta delta threshold cycle ($\Delta\Delta CT$)”

have not been published. These embryos should be removed from the analysis unless they were re-tested by a more comprehensive method.

We thank the reviewer for this comment. The PCR-24 test used in this study (initial validation described in (Treff et al., 2012, Treff and Scott, 2013)) was clinically validated by a number of studies including 2 randomized clinical trials that showed significant benefit from its use (Forman et al., 2013, Scott et al., 2013). More importantly, the laboratory performing the PCR-24 test utilized in this manuscript (Foundation for Embryonic Competence [FEC]) was certified by CLIA (Clinical Laboratory Improvement Amendments) in the United States and the test's quality control and quality assurance was signed off by New York and New Jersey States. Similar PCR-based tests have been designed and offered to IVF patients worldwide by a number of companies (including Reprogenetics, Cooper Genetics, iGenomix, among others). As correctly stated by the reviewer, most laboratories recently switched to NGS-based platforms for PGT-A. This transition was primarily due to the adoption of cryo-all protocols (cryopreservation of all embryos after biopsy for transfer during a subsequent cycle) by IVF clinics. Consequently, there was no longer a need for rapid, "in house" testing using a PCR-based approach, especially, as low cost and reliable NGS-based platforms became available.

Out of 93 embryos used for detailed post-implantation characterization, 82 were analysed by tNGS, 9 were analysed by PCR-24, and 2 were analysed by microarray. While we cannot re-test the trophoectoderm samples obtained at blastocyst stage (as different amplification strategies are used for PCR-24 and tNGS), only one out of 9 embryos analysed by PCR-24 at the blastocyst stage was re-analysed by tNGS on day 9, and the result was concordant with the blastocyst biopsy (embryo 104, euploid). These results indicate that the use of PCR-24 does not affect any of the analyses presented.

(6) Also, what was this the method used for the embryos that were mis-diagnosed? Was it 24 chromosome PCR?

The misdiagnosed embryos were sequenced both at blastocyst stage and day 9 by NGS. We now mention this in the text.

(7) How were the TSCs derived, blastocysts or first trimester placentas? No details are provided other than the culture conditions used.

We now provide the detailed information. We did not derive the human TSCs ourselves. As specified in our manuscript "CT human TSCs were kindly provided by Hiroaki Okae and Takahiro Arima (Tohoku University Graduate School of Medicine, Japan). The original paper in which the TSC derivation was described was cited in results, but we now also cite it in the methods section for clarity.

(8) The use of TSCs and hESCs for functional studies are not mentioned in the abstract.

As requested, we now mention this in the abstract.

(9) There are some grammatical issues. For instance, see "odds ratio of embryos with single chromosomal aneuploidy to have better blastocysts expansion".

(10) Differentiated is spelled wrong in Figure 5f.

We thank the reviewer for carefully reading our manuscript. We have corrected the spelling mistakes and the grammatical issues.

Reviewer #3 (Remarks to the Author):

Overview: In this study, Shahbazi et al., use their recently established in vitro method of culturing human embryos to characterize when and how aneuploid embryos fail during early post-implantation development. First, the authors analyzed a large number of blastocyst embryos tested by PGT-A and found that chromosomes 15, 16, 21 and 22 were most often affected by aneuploidies. Next, Shahbazi et al. confirmed previous findings and validated their culture system by showing that aneuploid embryos reaching the blastocyst stage at day 6 instead of 5, were highly likely to show defects at day 9. This criterion allowed them to focus only on aneuploid embryos reaching the blastocyst stage at day 5 as normal embryos do, and thereby clearly separate defects that arise after implantation from earlier deficiencies. The authors analyzed embryos carrying trisomy 21, trisomy 15, trisomy 16, monosomy 21 and euploid embryos as controls and concluded that: i) monosomy 21 embryos arrested largely by day 9, ii) trisomy 15 and 21 embryos developed normally until day 9, and iii) monosomy 21 and trisomy 16 embryos were significantly smaller due to hypoplastic trophoblast at day 9.

The authors then went on and identified mosaicism in 3 out of 29 embryos with monosomy 21, that accounted for a higher developmental potential of these mosaic embryos. Next, in an attempt to elucidate the mechanism behind the trophoblast hypoproliferation in trisomy 16 embryos, Shahbazi et al., performed some functional studies using human trophoblast stem cells as an in vitro model of trophoblast. They

concluded that both in the *in vitro* model and in the trisomy 16 embryos, increased levels of E-CADHERIN, a gene located on chromosome 16, result in increased differentiation and decreased proliferation of trophoblast cells.

General comments

The number of analyzed embryos in this study is very impressive and the overall characterization will provide the IVF community with an important resource. The general efficiency and quality of the human embryo analysis using the new embryo culture platform is at a high level.

We are grateful to the reviewer for his/her appreciation of the quality of our manuscript.

The identification of previously unidentified mosaicism was interesting but did not appear to be further relevant in explaining the phenotype seen in monosomy 21 embryos.

Our data indicates that approximately 40% (n=7) of monosomy 21 embryos arrest between day 7 and day 9 of development, whereas another 40% (n=7) of monosomy 21 embryos reaches day 9 with a normal morphology. We managed to re-sequence two day-9 monosomy 21 embryos, one from the first category and another from the second category. Interestingly, we observed that the embryo that had a normal morphology was a mosaic of WT and monosomy 21 cells, while the arrested embryo was fully aneuploid. Therefore, we can conclude that a proportion of the identified cases of monosomy 21 embryos that display a normal morphology on day 9 may be mosaic.

By far the most problematic aspect of the whole paper is the functional analysis using human trophoblast stem cells and the conclusions drawn from this. Firstly, the human trophoblast stem cells are *in vitro* representation of the villous cytotrophoblast, a cell type that appears later during development.

In our *in vitro* culture system of human embryos the villous cytotrophoblast appears during early post-implantation development and the same is true when other authors used this culture (Xiang et al., 2019, Zhou et al., 2019, Deglincerti et al., 2016, Shahbazi et al., 2016). Therefore, the human TSCs we used for our experiments are a valid model of the early post-implantation trophoblast. Nonetheless, we have overexpressed ECAD in a different trophoblast model. Following a protocol recently described (Mischler, 2019) we have converted human ESCs into early post-implantation-like human TSCs. New supplementary Figure 9e-g shows that upon ECAD overexpression converted human TSCs acquire a differentiated morphology and upregulate the expression of the differentiation marker HLA-G.

Secondly, Chromosome 16 contains about 800-900 genes (depending on the counting method), of which many are essential for cell viability and during development. Therefore, the hypothesis that the observed trophoblast phenotype is caused by an expected 1.5x upregulation of only one gene (CDH1) appears highly unlikely. However unlikely, this hypothesis could be tested in human trophoblast stem cells if the same gene dosage overexpression of expected 1.5x could be achieved. Unfortunately, the observed dox-inducible overexpression level of CDH1 seen in Supplementary Figure 8c is instead around 200-300 and therefore of a completely different order of magnitude. Consequently, it is completely unsound to compare these two models and draw any reliable conclusions from experiments in Figure 4 and Figure 5.

We agree and thank the referee for raising this valid point. To address this concern, we have performed doxycycline titration experiments. By modulating the concentration of doxycycline we have achieved a 1.6 fold upregulation of E-CAD in human TSCs. Under these conditions we still observe an upregulation of differentiation markers, a downregulation of stem cell markers, an increase in multinucleation, and a decrease in proliferation. All these new results are now presented on new Figure 5a to f.

Conclusion

This study provides new insights into post-implantation development of human aneuploid embryos and represents an important and useful resource for the IVF community. However, the functional analysis implicating E-CADHERIN as a cause of the trisomy 16 phenotype is deeply flawed and therefore inconclusive.

We appreciate the criticism of the reviewer and in response we have performed a comprehensive characterization of the role of E-CAD in trophoblast cells. We have upregulated E-CAD to the levels found in trisomy 16 embryos, we have performed loss of function studies, and we have validated our findings using a second TSC model. Our revised manuscript now presents strong molecular data in support of the role of E-CAD as a major contributor to the trisomy 16 trophoblast phenotype.

References:

- BIANCOTTI, J. C., NARWANI, K., BUEHLER, N., MANDEFRO, B., GOLAN-LEV, T., YANUKA, O., CLARK, A., HILL, D., BENVENISTY, N. & LAVON, N. 2010. Human embryonic stem cells as models for aneuploid chromosomal syndromes. *Stem Cells*, 28, 1530-40.
- DEGLINCERTI, A., CROFT, G. F., PIETILA, L. N., ZERNICKA-GOETZ, M., SIGGIA, E. D. & BRIVANLOU, A. H. 2016. Self-organization of the in vitro attached human embryo. *Nature*, 533, 251-4.
- FORMAN, E. J., HONG, K. H., FERRY, K. M., TAO, X., TAYLOR, D., LEVY, B., TREFF, N. R. & SCOTT, R. T., JR. 2013. In vitro fertilization with single euploid blastocyst transfer: a randomized controlled trial. *Fertil Steril*, 100, 100-7 e1.
- MISCHLER, A. K., V.; MAHINTHAKUMAR, J.; CARBERRY, C.; SAN MIGUEL, A.; RAGER, J.; FRY, R.; RAO, B. M. 2019. Two distinct trophectoderm lineage stem cells from human pluripotent stem cells bioRxiv.
- SCOTT, R. T., JR., UPHAM, K. M., FORMAN, E. J., HONG, K. H., SCOTT, K. L., TAYLOR, D., TAO, X. & TREFF, N. R. 2013. Blastocyst biopsy with comprehensive chromosome screening and fresh embryo transfer significantly increases in vitro fertilization implantation and delivery rates: a randomized controlled trial. *Fertil Steril*, 100, 697-703.
- SHAHBAZI, M. N., JEDRUSIK, A., VUORISTO, S., RECHER, G., HUPALOWSKA, A., BOLTON, V., FOGARTY, N. M., CAMPBELL, A., DEVITO, L. G., ILIC, D., KHALAF, Y., NIAKAN, K. K., FISHEL, S. & ZERNICKA-GOETZ, M. 2016. Self-organization of the human embryo in the absence of maternal tissues. *Nat Cell Biol*, 18, 700-8.
- TREFF, N. R. & SCOTT, R. T., JR. 2013. Four-hour quantitative real-time polymerase chain reaction-based comprehensive chromosome screening and accumulating evidence of accuracy, safety, predictive value, and clinical efficacy. *Fertil Steril*, 99, 1049-53.
- TREFF, N. R., TAO, X., FERRY, K. M., SU, J., TAYLOR, D. & SCOTT, R. T., JR. 2012. Development and validation of an accurate quantitative real-time polymerase chain reaction-based assay for human blastocyst comprehensive chromosomal aneuploidy screening. *Fertil Steril*, 97, 819-24.
- XIANG, L., YIN, Y., ZHENG, Y., MA, Y., LI, Y., ZHAO, Z., GUO, J., AI, Z., NIU, Y., DUAN, K., HE, J., REN, S., WU, D., BAI, Y., SHANG, Z., DAI, X., JI, W. & LI, T. 2019. A developmental landscape of 3D-cultured human pre-gastrulation embryos. *Nature*.
- ZHOU, F., WANG, R., YUAN, P., REN, Y., MAO, Y., LI, R., LIAN, Y., LI, J., WEN, L., YAN, L., QIAO, J. & TANG, F. 2019. Reconstituting the transcriptome and DNA methylome landscapes of human implantation. *Nature*, 572, 660-664.

REVIEWERS' COMMENTS:

Reviewer #1 (Remarks to the Author):

I thank the authors for adequately addressing the majority of my concerns. I do however have a few further remarks.

Based on the newly added text describing blastocyst quality, it is unclear whether the blastocyst grades included in the current analysis were assigned prior to thawing, after the thawing process, or prior to plating. The authors explain that they culture the embryos for an additional 24 hours following thawing. This may in fact affect the final blastocyst grade prior to extended culture, especially since the embryos have hatched. Please clarify the time of grading and ensure that the grades reflect the quality of the blastocysts just prior to plating. This is particularly important as the blastocysts are cultured in atmospheric oxygen after thawing, which is different to what is routinely used in IVF laboratories.

The culture of blastocysts for a further 24h after thawing also implies that embryos biopsied and frozen on day 6 are in fact plated on day 7 of development. The authors further explain that these embryos had poorer developmental potential during the extended in vitro culture. Could this be because these blastocysts are in fact plated later? Please clarify.

Reviewer #2 (Remarks to the Author):

In the revised version of their manuscript, Shahbazi and colleagues addressed the majority of my questions and concerns by performing additional experiments, including experimental details, and clarifying statements that did not necessarily support their findings. There are still, however, a few minor points that I would like to see be addressed as follows:

- (1) There are still grammatical issues with the manuscript. See "having a better blastocysts expansion score" on page 3, "albeit with different times" on page 4, and "differentiated populations are been specified" on page 11 as examples.
- (2) By the word, "Globally," do the authors mean "altogether" or "in general?" It's used quite a few times to summarize data, which seems like an overstatement as this also implies "worldwide."
- (3) In response to the use of PCR-24 for comprehensive chromosome screening, the authors state, "The PCR-24 test used in this study (initial validation described in (Treff et al., 2012, Treff and Scott, 2013) was clinically validated by a number of studies including 2 randomized clinical trials that showed significant benefit from its use (Forman et al., 2013, Scott et al., 2013). Again, I must point out that all of these studies are from the same group, which includes one of the authors on this manuscript, and has not been independently validated in a RCT. And while this test may be offered by genetic testing centers, it is not routinely used. Nevertheless, because only 9 embryos were assessed by PCR-24 and these embryos were not the ones that were mis-diagnosed, I am willing to overlook this.

Reviewer #3 (Remarks to the Author):

In the revised manuscript Shahbazi et al. have convincingly addressed this reviewer's concerns. The additional experiments and an independent validation using a second in vitro system provided further

support for the Authors' findings. In addition, the Authors took into account comments of Reviewer 1 and 2 and thereby significantly improved the clarity and quality of this manuscript. I therefore recommend publishing this work in Nature Communications.

REVIEWERS' COMMENTS:

Reviewer #1 (Remarks to the Author):

I thank the authors for adequately addressing the majority of my concerns. I do however have a few further remarks.

Based on the newly added text describing blastocyst quality, it is unclear whether the blastocyst grades included in the current analysis were assigned prior to thawing, after the thawing process, or prior to plating. The authors explain that they culture the embryos for an additional 24 hours following thawing. This may in fact affect the final blastocyst grade prior to extended culture, especially since the embryos have hatched. Please clarify the time of grading and ensure that the grades reflect the quality of the blastocysts just prior to plating. This is particularly important as the blastocysts are cultured in atmospheric oxygen after thawing, which is different to what is routinely used in IVF laboratories. The culture of blastocysts for a further 24h after thawing also implies that embryos biopsied and frozen on day 6 are in fact plated on day 7 of development. The authors further explain that these embryos had poorer developmental potential during the extended in vitro culture. Could this be because these blastocysts are in fact plated later? Please clarify.

Blastocysts were graded using Gardner's criteria just before they were frozen, as in clinical practice most transfer decisions are based on pre-freeze morphology. We then cultured the embryos for an additional 24 hours following thawing to give them full time to recover and hatch. We have now clarified this in the materials and methods section. As the reviewer mentions, at the time of plating, the embryos no longer have the same grades as when they were frozen; they expand and most of them hatch. In this case, Gardner's criteria are no longer helpful to assess the quality. We have noted the status of all the blastocysts 24 hours post-thawing as "hatched", "not hatched" and "dead/arrested" instead (Supplementary Table 2), and only plated embryos that hatched.

Reviewer #2 (Remarks to the Author):

In the revised version of their manuscript, Shahbazi and colleagues addressed the majority of my questions and concerns by performing additional experiments, including experimental details, and clarifying statements that did not necessarily support their findings. There are still, however, a few minor points that I would like to see be addressed as follows:

(1) There are still grammatical issues with the manuscript. See "having a better blastocysts expansion score" on page 3, "albeit with different times" on page 4, and "differentiated populations are been specified" on page 11 as examples.

We thank the reviewer for this comment. We have addressed these grammatical issues.

(2) By the word, "Globally," do the authors mean "altogether" or "in general?" It's used quite a few times to summarize data, which seems like an overstatement as this also implies "worldwide."

We thank the reviewer for this comment. We have used more precise language in place of "globally," such as "overall" and "in summary."

(3) In response to the use of PCR-24 for comprehensive chromosome screening, the

authors state, “The PCR-24 test used in this study (initial validation described in (Treff et al., 2012, Treff and Scott, 2013) was clinically validated by a number of studies including 2 randomized clinical trials that showed significant benefit from its use (Forman et al., 2013, Scott et al., 2013). Again, I must point out that all of these studies are from the same group, which includes one of the authors on this manuscript, and has not been independently validated in a RCT. And while this test may be offered by genetic testing centers, it is not routinely used. Nevertheless, because only 9 embryos were assessed by PCR-24 and these embryos were not the ones that were mis-diagnosed, I am willing to overlook this.

As we mentioned before, out of 93 embryos used for detailed post-implantation characterization, 82 were analysed by tNGS, 9 were analysed by PCR-24, and 2 were analysed by microarray. Only one out of 9 embryos analysed by PCR-24 at the blastocyst stage was re-analysed by tNGS on day 9, and the result was concordant with the blastocyst biopsy (embryo 104, euploid). Therefore, the use of PCR-24 does not affect any of the analyses presented.

Reviewer #3 (Remarks to the Author):

In the revised manuscript Shahbazi et al. have convincingly addressed this reviewer's concerns. The additional experiments and an independent validation using a second in vitro system provided further support for the Authors' findings. In addition, the Authors took into account comments of Reviewer 1 and 2 and thereby significantly improved the clarity and quality of this manuscript. I therefore recommend publishing this work in Nature Communications.

We thank the reviewer for his constructive criticism which helped us improve our manuscript.